

# 1 Properties of aerosols and formation mechanisms

# 2 over southern China during the monsoon season

**Weihua Chen[1], Xuemei Wang[2]\*, Jason Blake Cohen[3], Shengzhen Zhou[2]\*,**
**Zhisheng Zhang[4], Ming Chang[1], and Chuen Yu Chan[5]**
*[1] School of Environmental Science and Engineering, Sun Yat-Sen University,*
*Guangzhou, China*
*[2] School of Atmospheric Sciences, Sun Yat-Sen University, Guangzhou, China*
*[3] Department of Civil and Environmental Engineering, National University of*
*Singapore, Singapore*
*[4] South China Institute of Environmental Sciences, Guangzhou, China*
*[5] Key Laboratory of Aerosol, SKLLQG, Institute of Earth Environment, Chinese*
*Academy of Sciences, Xi'an, China*
[*]Corresponding author:
Xuemei Wang (eeswxm@mail.sysu.edu.cn)
Shengzhen Zhou (zszking@126.com )



**Abstract**
Measurements of size-resolved aerosols from 0.25 to 18 μm were conducted at three
sites (urban, suburban and background sites) and used in tandem with an atmospheric
transport model to study the size distribution and formation of atmospheric aerosols in
southern China during the monsoon season (May-June) in 2010. The mass
distribution showed the majority of chemical components were found in the smaller
size bins (<2.5 μm). Sulfate, was found to be strongly correlated with aerosol water,
and anti-correlated with atmospheric $SO_2$, hinting at aqueous-phase reactions being
the main formation pathway. Nitrate was the only major species that showed a
bi-modal distribution at the urban site, and was dominated by the coarse mode in the
other two sites, suggesting that an important component of nitrate formation is
chloride depletion of sea salt transported from the South China Sea. In addition to
these aqueous-phase reactions and interactions with sea salt aerosols, new particle
formation, chemical aging, and long-range transport from upwind urban or biomass
burning regions were also found to be important in at least some of the sights on some
of the days. This work therefore summarizes the different mechanisms that
significantly impact the aerosol chemical composition during the Monsoon over
southern China.
**Keywords:** chemical component, mass size distribution, aqueous-phase reaction
chloride depletion



## 1. Introduction

Atmospheric aerosols are solid and liquid substances ubiquitously suspended in
the Earth's atmosphere, that impair visibility, negatively affect human health, and
directly and indirectly impact regional and global climate (Chung and Seinfeld, 2005;
Cohen et al., 2011; Jacobson, 2001; Kim et al., 2008; Ramanathan and Carmichael,
2008; Rosenfeld et al., 2014; Tao et al., 2009; Burnett et al., 2014). The size
distributions and chemical composition of aerosols play essential roles on their
transport, transformation, removal mechanisms (Seinfeld and Pandis, 2006; Zhao and
Gao, 2008a; Giglio et al., 2003, 2006; Cohen and Wang, 2013; Petrenko, et al., 2012;
Cohen and Prinn, 2011; Delene and Ogren, 2002; Dubovik et al., 2000). And also, to
some extent, they provide useful information to validate and improve model
performance (Pillai and Moorthy, 2001; Cohen and Wang, 2013; Myhre et al., 2013;
Schuster et al., 2006; Tsigaridis et al., 2014; Cohen and Lecoeur, 2015; Cohen, 2014;
Cohen and Wang 2013). In the environment, the most important aerosol processes
occur over the aitken, condensation, droplet, and coarse size modes, where new
particles form in the condensation mode, and in-cloud processing and aqueous
reactions occur in the droplet mode (Yao et al, 2003a; Meng and Seinfeld, 1994;
Wang et al., 2012; Volkamer et al., 2009; Lim et al., 2010; Ervens et al., 2011). On the
other hand, coarse mode aerosols are usually due to different source types and
therefore provide further information about the aerosol distribution at a given
location.
Previous research suggests that sulfate is mostly contained in the non-coarse



modes, with the conversion of $SO_2$ occurring mostly via gas-phase oxidation followed
by condensation, or through droplet mode sulfate produced from fog/cloud process
(Meng and Seinfeld, 1994; Barth et al., 1992). On the other hand, nitrate usually has a
bi-modal distribution with peaks in both the fine and coarse modes. Fine mode nitrate
is formed mainly by oxidation of $NO_2$ to $HNO_3$ and subsequent condensation, or from
the heterogeneous hydrolysis of $N_2O_5$, while coarse mode nitrate is often observed
due to the effect of chloride depletion of sea salt aerosols (Pierson and Brachaczek,
1988; Harrison and Pio, 1983). Ammonium is mostly found in the fine mode and is
chemically associated with sulfate and nitrate. Carbonaceous materials, organic
carbon (OC) and elemental carbon (EC), are both found primarily in the non-coarse
mode. While both OC and EC are impacted by differing emissions sources and wet
deposition, there are other significant differences: EC is hydrophobic and radiatively
active, while OC is hydrophylic and further has significant source terms from
condensation and secondary particle formation (Lan et al., 2011).

Meteorological conditions also play a vital role in the size distribution and the

formation of secondary aerosols. Southern China has high relative humidity and
temperature, leading to significant aerosol water uptake and secondary aerosol
formation and processing. Furthermore, during the Monsoon period, South China is
greatly affected by air masses transported from the South China Sea, leading to a
large variation in the upwind aerosol compositions and loadings as compared to those
from local or continental sources.

In this paper, we present a unique database of the size-different mass distribution





of sulfate, nitrate, ammonium and carbonaceous aerosols during the Monsoon Season
over southern China. The data is sampled from a combination of three different sites,
one in an urban area, one in a suburban area, and one in a remote area, providing
further insights into the characteristics in each of these regions. The measurements
made during the observation periods were analyzed in tandem with each other and a
meteorological model, leading to some robust conclusions regarding the formation
mechanisms of water soluble ions, the identification and impacts of long-range
transport of biomass and urban sources, and the impacts mixing sea-salt and urban
pollutants.

**2. Measurements and methodology**
**2.1. Description of the sampling sites**
The field study was conducted at three sites in southern China (Figure 1), two of
which were situated in Guangdong and the other in Hainan. Guangdong is located in a
subtropical monsoon climate, primarily influenced by cold and dry air masses from
the North in December to February, and warm and wet air masses from the South
China Sea in May to August. It has a single annual local rainy season extending from
April to September. Hainan is located further to the south, and has year-round warm
to hot weather and a distinct rainy season from May to October.
The first site was set at (23.12 N, 113.36 E), on the rooftop of a building in the
South China Institute of Environmental Sciences, Guangzhou (GZ), an urban
mega-city containing more than 13 million people. The site was located about 50m





above ground, in an area surrounded by residential and commercial buildings, with
the nearest arterial roads located about 200m away. There were no significant
industrial emission sources found around the site. This site was chosen since it is
highly representative of a typical megacity.

The second site was located at (22.34 °N, 113.58 °E), on the rooftop of the library

at Sun Yat-Sen University, in the city of Zhuhai (ZH), a medium sized city of about
1.6 million people located in Southern Guangdong adjacents to Macau. The site was
located about 60m above the ground, in an area surrounded by mountains on three
sides and the estuary where the Pearl River meets the South China Sea about 500m
away on the fourth side. There are no significant industrial or major transportation
emissions sources nearby. This site was chosen since it is highly representative of a
coastal partially urbanized area.

The third site was located at Jianfeng Mountain (JFM, 18.74 °N, 108.86 °E), in a

tropical rainforest situated at the Southwest corner of Hainan. This site is distant from
the major cities of Hainan province and is further located about 5km away from the
coast. JFM is not directly influenced by anthropogenic emissions and is generally
regarded as a background site to investigate the long-rang transport (Zhang et al.,
2013a). This site was chosen both because it is representative of a remote site and
because it receives air masses from three different directions: continental East Asia to
the North, the South China Sea to the South, and Southeast Asia to the West.

**2.2 Sampling of aerosol**



The sampling campaign was performed in May and June 2010. To attain
size-segregated particle samples, a 6-stage High Flow Impactor (MSP) with an
airflow rate of 100 L min$^{-1}$ was employed, with cutoff diameters ($D_p$) of 18, 10, 2.5,
1.4, 1.0, 0.44 and 0.25 μm. A total of 6 sets of size-segregated particle samples were
collected on 77 mm and 90 mm (for inlet) quartz microfiber filters (Pall Corporation,
NY, USA). 24h sampling was performed every other day in GZ and ZH, while 48h
sampling was conducted every day in JFM. Over the entire duration of the campaign,
there were 70 samples taken in GZ, 56 samples taken in ZH, and 140 samples taken in
JFM. Detailed information of the aerosol sampling and in-lab chemical analytical
techniques can be found in Zhang et al. (2013a).

To be consistent with the background literature and the constraints of the size

bins measured in this study, we implement 2.5 μm as the cut-off size to separate fine
and coarse particles, and the size bins from 0.44-1.4 μm to define droplet particles.

Although we were not able to directly measure aerosol water content, given its

importance for the study here, we instead to estimate the amount by the use of AIM-II
model (Clegg et al., 1998). Implementation of the model required the use of measured
molar concentrations of sulfate [$SO_4^{2-}$], nitrate [$NO_3^-$], ammonium [$NH_4^+$], ambient
temperature (T), and relative humidity (RH). Further, an approximation of the particle
strong acidity [$H^+$]$_s$. is required, which also has been computed from measurements
following Eq. (1).
$$\left[ H^+ \right]_s = 2\left[ SO4^{2-} \right] + \left[ NO_3^- \right] - \left[ NH_4^+ \right] \tag{1}$$






### 2.3 Meteorological data

Meteorological parameters, including wind speed (WS), wind direction (WD),

temperature (T), relative humidity (RH), pressure (P), and precipitation were

simultaneously monitored in GZ and JFM sites with a time resolution of 30 minutes.

The same meteorological parameters in ZH, as well as the daily low-level cloud cover

data at all three sites, were obtained from the China Meteorological Data Sharing

Service System (http://data.cma.cn/site/index.html).

### 2.4 Remotely sensed measurements

Aerosol optical depth (AOD), Fire Radiative Power (FRP), and Fire Quality

Assurance [QA] data were obtained from the MODIS sensors aboard both the AQUA

and TERRA satellites. Specifically, we obtained the Collection 6, 3km Level 2 swath

product for AOD [Remer et al., 2013], and Collection 5.1, 1km Level 2 swath

products for FRP and QA[Giglio et al., 2006]. All of the data is cloud-screened, with

AOD data being computed using different algorithms over land and water, and the fire

data using 19 different channels for quality assurance. We only accept values for FRP

and Fire Count where the QA is at least 90%.

### 2.5 Atmospheric transport model

Two Lagrangian particle dispersion models, the Hybrid Single Particle Lagrangian

Integrated Trajectory (HYSPLIT) (Draxler and Hes, 1998) and FLEXPART coupled

with The Weather and Research and Forecasting (WRF) model were used to compute





air parcel trajectories (Stohl et al., 1998; Brioude et al., 2013). HYSPLIT uses single
air parcels to compute trajectories with the use of Global Data Assimilation System
(GDAS, $1°\times1°$) as input data. FLEXPART, on the other hand, uses a larger number of
air parcels to compute trajectories based on the meteorological predictions provided
by mesoscale model WRF

An Eulerian model, WRF/Chem V3.4.1 was used in this study to simulate fog

processing. For this mode, the target region's was modeled at a spatial resolution of 3
$\times$3km. Detail information about the WRF/Chem model set-up refers to Situ et al.
(2013). While WRF was used to simulate the meteorological fields required for the
FLEXPART back trajectory calculations over the larger region. In this case, the
region was modeled with a spatial resolution of $27\times27$ km and a temporal
resolution of 1 hour.

**3.  Results and discussion**
**3.1. Overall aerosol characteristics**

The mass time series of the total aerosol mass ($PM_{10}$) at the three sites has an

average and standard deviation of 46.7±20.6, 23.7±7.3, and 8.0±2.6 μg m$^{-3}$ in GZ, ZH,
and JFM respectively (Figure S1). The mean and range of $PM_{10}$ in highly urban GZ
was both higher and wider than in suburban ZH and rural JFM, with the respective
ranges being [22.5, 92.3], [12.9, 34.6], and [4.6, 14.2] μg m$^{-3}$ in the three sites. In
terms of the mass size distribution, the percentage of $PM_{1.0}$ to $PM_{10}$ and $PM_{2.5}$ to
$PM_{10}$ fell within the range of [0.52, 0.55] and [0.72, 0.76] respectively (Table 1).





When considered as a whole, it is the smaller sized particles that dominate the aerosol
loading at all three of these sites.

Looking at the data on a species-by-species level, the majority of individual

chemical species contribute at least 57% to $PM_{2.5}$. The sole exception is nitrate at ZH
and JFM, which were mainly concentrated in the coarse mode with a percentage of
above 90%. Overall, the sum of five major chemical components (i.e. sulfate, nitrate,
ammonium, OC, and EC) accounted for about 90% of the total mass concentration of
detected chemical components across all three sites.

Two of the species, sulfate and OC, were found to dominate particle composition,

with concentration of $11.7 \pm 5.2$, $8.8 \pm 3.2$, $2.2 \pm 1.5$ $\mu g\ m^{-3}$ for sulfate and $7.2 \pm 2.7$,
$3.0 \pm 1.5$, $1.8 \pm 0.8$ $\mu g\ m^{-3}$ for OC in GZ, ZH and JFM, respectively. Sulfate
concentration was much higher than that of OC in urban and suburban locations no
matter what the particle size was, while OC concentration was similar to that of
sulfate in fine particles and slightly higher in coarse particles at the remote site. These
findings are consistent with the nature of the sources of sulfur from industrial and
shipping sources.

Nitrate, although primarily formed similar sources as sulfate, such as mobile

vehicles and high temperature industry, showed a remarkable difference between
urban and background site, with ranging from fourteen to thirty times higher in GZ
than in the other sites, especially for fine mode nitrate. This is consistent with its more
rapid oxidation of its precursor species, especially so in the urban atmosphere (Cohen
et al., 2011). Furthermore, it was found to have a relatively insignificant concentration





in ZH and JFM, indicating far less anthropogenic emission of the precursor over these
two sites.

The values of OC and EC in $PM_{2.5}$ were $7.2 \pm 2.7$ and $3.4 \pm 3.2$ $\mu g\ m^{-3}$ in GZ,

$3.0 \pm 1.5$ and $1.5 \pm 0.9$ $\mu g\ m^{-3}$ in ZH. These values were lower than that of found in
previous studies done in GZ and ZH during the wet season: OC and EC were 13.1 and
4.6 $\mu g\ m^{3}$ in GZ in 2007, 14.8 and 8.1 $\mu g\ m^{-3}$ in GZ in 2002, and 5.4 and 1.9 $\mu g\ m^{-3}$ in
ZH in 2002 (Cao et al., 2004; Tao et al., 2009). Furthermore, OC and EC
concentrations in JFM were found to be lower than that at other forest sites in China,
such as Hengshan: 3.01 and 0.54 $\mu g\ m^{-3}$ in 2009 (Zhou et al., 2012), Daihai: 8.1 and
1.81 $\mu g\ m^{-3}$ in 2007 (Han et al., 2008), and Taishan: 6.07 and 1.77 $\mu g\ m^{-3}$ in 2007
(Wang et al., 2011). However, the EC and OC in JFM were similar to some
background sites in other countries, such as Puy De Dome in France: 2.4 and 0.26 $\mu g$
$m^{-3}$ in 2004 (Pio et al., 2007) and Sonnblick in Austria: 1.38 and 0.23 $\mu g\ m^{-3}$ in 2003
(Pio et al., 2007). This finding is not unexpected, since there are very few urban
sources near the site. It is therefore relatively representative of a remote background
site, and will be treated as such subsequently in this paper.

**3.2. Size distribution by chemical composition**

The mass size distribution of major compositions at the three sites during the

study period, showing that sulfate had a single-peaked distribution, with the
maximum value found in the 0.44-1.0 $\mu m$ size over all sites and under all different
meteorological conditions examined in this study. The droplet mode sulfate was about



56.0 ± 8.0 %, 63.5 ± 5.1 % and 58.8 ± 9.4 % of the total sulfate mass in GZ, ZH and
JFM, respectively (Figure 2). This confirms that secondary processing is essential,
with aqueous-phase reactions playing a crucial role on the formation and/or growth of
droplet sulfate, throughout all of these different regions. It is interesting to note that
ZH had the highest relative concentration of droplet model sulfate, which although it
is less urban than GZ, is consistent with the fact that it is located very close to large
amounts of sulfur emissions from the shipping traffic at the massive nearby ports of
Hong Kong and Shenzhen.
Droplet mode ammonium was mainly due to ammonia vapor that reacted with or
condensed on an acidic particle surface. Ammonia was observed to highly correlate
with sulfate at the three sites (R>0.81, P<0.01), particularly so in the size range of
0.44-1.0 μm. This is consistent with the fact that sulfuric acid preferentially reacts
with ammonia (Zhuang et al., 1999), and that most of sulfate in the atmosphere is
generally found as ammonium sulfate in the droplet mode (Liu et al., 2008; Zhuang et
al., 1999).
The nitrate size distribution was found to be bi-modal in GZ, with the peaks
occurring in the 0.44-1.0 μm and 2.5-10 μm size ranges. However, it was found that
the majority of nitrate was found in the fine mode particles when the air came from
continental sources with the percentage of 33%, and conversely it was found in the
coarse mode particles when the air came from an oceanic source with the percentage
of 51%. This is consistent with the fact that droplet mode nitrate is formed similarly
to sulfate, after oxidation of the NOx, but is only converted into aerosol after all of the





sulfate first reacts, and only in the presence of sufficient ammonia (Zhuang et al.,
1999). On the other hand, this result is consistent with the fact that nitrate was found
mostly in the coarse mode in ZH and JFM, where it accounted for up to 40% of total
particulate mass. A higher relative humidity, consistent with the warm and wet
atmosphere over the South China Sea, makes gaseous nitric acid more likely to be
absorbed by coarse particles in the atmosphere (Anlauf et al., 2006), resulting in a
higher relative concentration of nitrate in the coarse mode in ZH and JFM (where the
relative humidity averaged 80 and 91% respectively, as compared to only 73% in GZ).
Further, the presence of coarse mode nitrate is consistent with chlorine reduction, as
talked about later.

OC and EC showed a similar mono-modal distribution in GZ and ZH, with a

dominant and broad peak over the range from 0.25-1.4 μm. On the other hand, a
bi-modal distribution was found in JFM. In urban and suburban areas, there are
significant primary sources from traffic and industry in the e.g. Huang et al. (2006)
and Cao et al. (2004). It is also consistent with the high levels emissions due to the
ship traffic to Shenzhen and Hong Kong, both of which are located near ZH, which in
turn would compensate for the otherwise reduced industrial and traffic sources. OC
has both primary sources, which are similar to those for EC as well as secondary
formation. There were a few days in which the ratios of OC to EC are not consistent,
indicating a large secondary source of OC. We investigate these days and find that
long-range transported of far-upwind urbanization and biomass burning is responsible,
as talked about later. Additionally, there is some coarse mode OC present in JFM,



suggesting a possible source of biological aerosol, which is consistent with the large
amounts of vegetation present in that region.

**3.3. Observed Aqueous-phase reaction of droplet mode sulfate**
The daily droplet mode sulfate ranged from 3.0-13.6, 1.6-9.5 and 0.5-4.9 $\mu g\ m^{-3}$
in GZ, ZH and JFM respectively. The cases with concentration of droplet sulfate
above the mean plus one standard deviation ($8^{th}$ and $12^{th}$ May in GZ, $12^{th}$ May and $1^{st}$
Jun. in ZH, and $4^{th}$ and $13^{th}$ May 2010 in JFM) were chosen to investigate the effect
of aqueous-phase reaction in the formation of droplet mode sulfate (blue shade in
Figure S1). In each of these cases, it was found that droplet mode sulfate accounted
for about two thirds of the total mass concentration of sulfate at the three sites,
indicating that the average size was small and that the particles were therefore
relatively young, strongly indicative of new particle formation.
A backward trajectory analysis found that during these events, the air masses at
these sites mainly originated over the South China Sea (figures not show here).
Additionally, it was determined that during these times at the sites there was an
abnormally high amount of low cloud cover 60-70% and a relatively higher relative
humidity (75~83%) (Table 2). This combination is consistent with moist air being
transported over land where ship and industrial $SO_2$ emissions can undergo chemistry
in the presence of large amounts of liquid cloud water, to form droplet-model sulfate.
We estimated the liquid water content using the AIM-II model (Equation 1). The
results showed a significant correlation with droplet mode sulfate in GZ (R=0.98,





P<0.05), ZH (R=0.53, P<0.05) and JFM (R=0.80, P<0.05), indicating that water
content correlated closely with the sulfate aerosol loadings. This is further evidence
that aqueous formation was likely an important contributing factor.

We further investigated the aqueous-phase reaction of particles due to fog

processing for the data from 8[th] May in GZ. This is because the measured visibility
met the World Meteorological Organization cutoff value of less than 1 km due to
water droplets, in the early morning (05:00-07:00 LT) (Figure 3(c)). Consistently,
during this time, it was found that the relatively humidity was quite high (RH>90%)
and the wind was quite low (wind speeds<1.0 m s$^{-1}$). Also during this time, the cloud
fraction and simulated 2m relative humidity were up to 90% over Southern China
(Figure 3(a-b)). Furthermore, the depression dew point ($\triangle T=T-Td$, while Td denotes
dew point temperature) was lower than 1 (Figure 3(c)), which indicating that vapor
pressure was saturated. An accompanying analysis using WRF/Chem of the simulated
cloud water mixing ratio was the highest during this period over the GZ area and
higher value was found around 06:00 LT (Figure 3(e-f)). This combination promoted
the existence of fog/low cloud.

Further analysis was done by looking at measurements of $SO_2$ (data from

Guangzhou Environmental Protection Bureau, http://www.gzepb.gov.cn/ ). The
diurnal variation on 8[th] May showed a unique pattern compared with the mean diurnal
pattern as measured during 2009-2011(Figure 3(d)). On this day, the $SO_2$
concentration decreased dramatically from 05:00-07:00 LT, which is consistent with
$SO_2$ transferred from gas to aqueous phase due to the high solubility of $SO_2$ in fog





water droplets (Zhang et al., 2013b).

Simulation of these conditions using WRF/Chem indicates that rapid growth of

both Aitken and accumulation mode sulfate started at 07:00 LT and peaked at
08:00-09:00 LT (Figure 3(g-h)). This further supports the conclusion of fresh sulfate
production, in this case through both the aqueous and potential initial gas to particle
formation, followed by condensation/coagulation and uptake into the liquid droplets
present. All of this is consistent with generalized urban modeling studies performed
under similar conditions (e.g. Cohen and Prinn (2011]).

**3.4. Observed interactions between nitrate and chloride depletion**

The mass size distribution of sodium and chloride showed a similar pattern to

nitrate at the three sites, peaking in coarse mode particles (Figure 4) with an average
percentage of 43%, 62% and 43% for coarse mode sodium, 53%, 76% and 74% for
coarse mode chloride in GZ, ZH and JFM, respectively. The percentage of chloride
depletion (%$Cl_{dep}$) (Figure 5) was calculated using Eq. (2), where [$Cl_{meas}^-$] and
[$Na_{meas}^+$] are the measured equivalent concentrations of chloride and sodium
respectively [Yao et al., 2003b].

$$\%Cl_{dep} = \frac{1.174\left[Na_{meas}^+\right] - \left[Cl_{meas}^-\right]}{1.174\left[Na_{meas}^+\right]} * 100\% \qquad (2)$$

In general, the %$Cl_{dep}$ decreased as the aerosol mass increased. The relationship

was strongly pronounced for fine mode sea salt particles, having a significant
relationship between sodium and chloride at the three sites (R= [0.50, 0.61], P<0.05).
On the other hand, there was no statistically significant correlation found in the coarse



particles. Chloride had been almost entirely depleted in fine mode particles with only
53% and 31% depleted in fine mode particles in ZH and JFM, while there was 89%
and 91% depleted in coarse particles in ZH and JFM. The result is consistent with a
study conducted in South China Sea in 2004 as well as theory that reaction between
sulfuric acid and nitric acid with sea salt (sodium chloride) is facilitated in fine
particles due to their larger surface areas to volume ratio (Chatterjee et al., 2006; Hsu
et al., 2007).

The ratio of calculated ammonium to measured ammonium was used to explain

the presence of sulfuric acid and nitric acid in the aerosol, with a value larger than 1
indicating there was insufficient ammonium to neutralize nitric acidic $NO_3^-$ (since
ammonium first consumes sulfuric acid). The calculated ratio was much higher than 1
in ZH and JFM suggesting that nitrate plays a role in Cl depletion. The ratio of nitrate
to percent chloride depletion can then be used to calculate the contribution of coarse
nitrate to chloride depletion (Zhuang et al., 1999; Zhao and Gao, 2008b). This result
showed that nitrate was responsible for the depletion of 54% and 17% of coarse
chloride in ZH and JFM respectively. This suggests that the interaction of sea salt
particles with anthropogenic pollutants is an important pathway for the generation of
aerosol species in coastal suburban regions like ZH, which have sizable amounts of
both sea salt and $NO_x$ emissions.

Furthermore, we analyzed the chloride depletion rate in coastal ZH and JFM

under different air masses conditions, and found the total chloride depletion was 88.0%
and 53.5% when the air masses came from the ocean, as compared with 91.2% and



53.8% when the air masses came from the continent. In general, the mean RH was
82.5% when the air masses came from the ocean, while the RH was 78.3% when the
air masses came from the continent. The consistent finding is that there was a higher
percentage of chloride depletion found when the air was relatively less humid,
suggesting another important non-linear effect between maritime aerosols
anthropogenic NOx (Chatterjee et al., 2010; Liu et al., 2008).

Relative humidity exceeded 80% during the whole sampling time in ZH except

for 24th May, which was 64%. The percentage of chloride depletion was 95% and 69%
in fine and coarse particles on 24th May, respectively. The only other day which had a
significant continental wind source at ZH also had a higher relative humidity (80%),
on 7th June. On that day the percentage of chloride depletion was 78% and 64%
respectively. While there was no distinct difference found in coarse particles for the
two cases, there was a considerable difference in the chloride depletion of the fine
particles. This finding is consistent with our understanding of the release of
hydrochloric acid under the known high nitric acid conditions, especially when there
is less aerosol water (at lower relatively humidity) to dissolve all of the volatiles, as
already discussed in the sections above (Chen et al., 2013; Dasgupta et al., 2007).

**3.5. The effects of long-range transport, and in-situ chemistry**

There were four days that the amounts and properties of the aerosols were

significantly impacted by long range transport and unique formation and alteration



mechanisms: one in each GZ and ZH (both occuring on 12$^{th}$ June) and three in JFM
(1$^{st}$, 3$^{rd}$, and 5$^{th}$ June).

On 12$^{th}$ June in both GZ and ZH, the total aerosol concentration was the highest

measured, at respectively 93.7 and 35.1 μg m$^3$ in GZ and ZH (Figure S1). Secondly,
the concentration of secondary soluble ions was the highest measured, in GZ based on
the Sulfur Oxidation Ratio and Nitrogen Oxidation Ratio (Sun et al., 2006), with the
respective values being 0.20 (SOR) and 0.17 (NOR) over the 0.44-1.0μm (Figure 6
(a-b)) (no supported data to estimate SOR and NOR in ZH on this day),. Thirdly, this
was the only day in GZ that the nitrate size distribution was found to be uni-modal,
where it peaked in the 1.0-1.44μm size range (Figure 6 (d)), which was the largest of
any mean size nitrate in GZ measured. Meanwhile, the nitrate size distribution
changed from coarse mode to bi-modal and peaked in 0.25-0.44 μm size range in ZH
measured on this day (Figure S2(j)). Fourthly, the peak of sulfate and ammonia
shifted from typical values in the 0.44-1.0 μm size range to the 1.0-1.44 μm size range
(Take GZ for example, Figure 6(c) and Figure S2(f)). All of these are consistent with
enhanced secondary production. Such a statistically enhanced amount of secondary
production requires the aerosols to have had considerably more time in the
atmosphere to have aged as they have, and therefore is consistent with them having
undergone considerable long range transport (Cohen et al., 2011).

To provide further evidence, 3-day air mass backward trajectories were conducted

at each of the three sites. The parcels were released at initial heights of 100, 500,
1000 and 2000m, hourly, as a means of robustly sampling the boundary layer





throughout the day. The results showed that air masses winded up over GZ and ZH
in the lower free troposphere or near the top of the boundary layer had mostly
originated over continental Southeast Asia, while those winding up near the surface,
had mostly come from Northern China (Take GZ for example, Figure S2(a)).
Furthermore, since the air masses came from opposite directions at nearly the same
time, the end result was observed to be a stable meteorological condition over GZ
(very low wind 0.1 m s$^{-1}$) and ZH (wind speed was 1m/s which was the lowest
during the sampling times). In fact, it seems from the back trajectory analysis that
there was descending air in and around GZ and ZH on this day, which implies that
air transported from far away in the lower free-troposphere would have been
transported back near the surface (Take GZ for example, Figure S2 (b-c)). All of
these results were further consistent with the high levels of aerosols measured as
well as additional secondary processing having had time to occur.

FLEXPART-WRF was next applied to address the issue of the air residence time

through the column over GZ and ZH on that day. Take GZ for example, as can be
shown in Figure 6 (e-f), there was a strong influence from the region local to GZ and
surrounding adjoining cities, at a lower altitude (500m and lower) (Figure 6(e) and
Figure S2 (d-e) ). Also, the results showed that air from far away was contributing
mostly to the residence time at higher altitudes, yet still in the boundary layer, at
1000m and above (Figure 6(f)). This is further evidence that indeed long-range
transport was also responsible.

Furthermore, the ratio of OC to EC concentrations was the minimum measured



values on the 12th June, with a mean ratio of 1.32 and 2.39 in GZ and ZH,
respectively . Also, OC showed a bi-modal distribution, although predominantly in
the fine mode while EC mostly peaked at fine mode particles (Take GZ for example,
Figure S2 (g-h)), indicating that the organic aerosol was mostly primary, as would be
expected from large fire sources. Additionally, the potassium concentration on the 12th
June was about 2-3 times higher than that of mean value measured in GZ and
ZH(Take GZ for example, Figure 10(a-b)) All of these findings above, including the
time of the year and the location, are consistent with the existance of biomass burning
over Southeast Asia being a likely source (Cohen, 2014).

At JFM the total aerosol concentration was highest on the 1st, 3rd, and 5th June. In

particular, the levels of EC and potassium were elevated on all three days, and the
ratio of OC to EC was depressed (Figure S3 (c-e)). However, in addition to these
clues, there were some differences: the levels of sulfate and ammonia were
remarkably elevated on the 3rd and 5th June (Figure 7(g) and Figure S3 (b)), likely due
to a mixing of urban sources with the fire sources. On the other hand, on the 1st June,
the sulfate was lower, but the nitrate was considerably higher, peaking in the coarse
mode (Figure 7 (h)), likely due to mixing of South China Sea air with the fire sources.

HYSPLIT results showed that on all three of these days, the great majority of air

masses arriving at JFM originated from continental Southeast Asia (Figure S3(a).
However, all of these parcels of air arrived in the upper boundary layer or the lower
free troposphere. By analyzing the FLEXPART-WRF runs at higher resolution, it was
demonstrated that there was a strong influence of air from ocean on 1st June (Figure 7





(a-b)) at the lower parts of the boundary layer. This is consistent with the observed
non–elevated sulfate and elevated coarse nitrate on that day. Furthermore, the
FLEXPART-WRF runs at higher resolution demonstrated a considerably influence of
air from Southern China (urban and semi-urban Guangdong Province, including many
major shipping lanes) on the 3$^{rd}$ and 5$^{th}$ June, again in the lower parts of the boundary
layer (Figure 7(c-f)). This is again consistent with the observed elevated levels of
sulfate, due to the in-situ processing of urban emissions as the air was transported to
JFM, and then mixing with the fire emissions transported from the other direction at
height. Additionally, there was some amount of fine mode nitrate found on the 3$^{rd}$ Jun.,
further consistent with the in-situ processing of NO$_2$ emitted along with biomass
combustion, and therefore further evidence that mixing occurred between the two
different source regions.

**3.6. Quantifying the impacts of fires**

Taking a first look at the possibility that fires are responsible, as described above,

we look at a summary of the statistics of the MODIS Fire Hotspots (Figure 8). As we
observe, while the total number of fire hotspots occurring throughout Southeast Asia
is moderate in early May, the number reduces to the extent that there are effectively
almost no burning parcels. Furthermore, those few square kilometers that are burning
are of low radiative intensity, under 200W/m$^2$, and hence only moderately or lowly
emitting, with the exception of a single day in late June, after the period of interest
has ended. This result shows that the fires themselves are not very important, or are



mostly obscured, which is consistent with previous findings over the region of both
high cloud cover and a large number of small or otherwise hard to detect fires (Cohen,
2014; Giglio, 2006).

Instead, we follow the approach of Cohen (2014) and instead look at the once to

twice daily measured AOD data (Figure 9), in the context of the Empirical Orthogonal
Functions approach. The rationale is that over Southeast Asia there are only a few
known large urban centers (Hanoi, Ho Chi Minh City, and Bangkok). Therefore, any
other significant contribution to the variance of measured AOD must be from fires.
The EOF technique has been shown to be an optimal manner by which to reproduce
both the spatial extent of and magnitude of the smoke over Continental Southeast Asia
(Cohen, 2014; Cohen and Leocure, 2015).

As observed, the major regions of high AOD (average AOD > 0.4) are found over

Southeast Asia as described above, with most of the sources coming from fires found
in two arcs: one from Eastern Thailand, through Laos, and ending in Central Vietnam;
and the other in the forests of Myanmar. The region around Hanoi is hard to descipher,
as it could be urban expansion or fire. Additionally, there are regions found in urban
East Asia, including the region between Hong Kong and Guangzhou and urbanization
along the Yangtze River, however, all of these are known regions of urbanization and
are not regions where fire is important (Figure 10).

An EOF Analysis concludes that in fact these are the only two statistically

significant EOFs. The measured AOD over both of these regions is clearly elevated
compared with the region as a whole throughout the entire time. Furthermore, there is





an especially large contribution from these two EOFs compared with the background
over Southeast Asia only (excluding AOD measured over China, which is downwind
and hence not a fire source region) from May $31^{st}$ to June $6^{th}$. Given the rapid
transport time from Southeast Asia to JFM, the fact that these peaks occur within 1
day of the peaks in the fires is reasonable. Additionally, while the overall Southeast
Asian AOD drops from the $8^{th}$ onwards, there is a very significant difference
(difference in AOD more than 0.5) between the overall AOD and that over the two
source regions again from June $8^{th}$ to June $13^{th}$. Given that there are markers of fires
in GZ and ZH on June $12^{th}$, including high potassium and a low OC/EC ratio, and that
a significant portion of the airflow over these regions originated from Southeast asia
within the past 72 hours, these results are consistent with high fires originating from
Southeast Asia then being transported over the next 72 hours to GZ and ZH. The fact
that only one day has such measured conditions at the surface is likely due to the fact
that the smoke is mostly concentrated near the boundary layer and hence local vertical
mixing was most prevelant on or around June $12^{th}$.

**4. Conclusion**
Aerosol samples were collected at three sites using a 6-stage sampler during the
local wet season in Southern China (May – Jun.) in 2010, to jointly study the mass
and size distributions of aerosol chemical components. Based on specific case studies,
some models of the air flow, and remote sensing, the impacts of chemistry and
atmospheric transport were investigated on the aerosol formation mechanisms at the



three sites over Southern China. These were chosen such that they spanned different
source and meteorological regions, at urban site GZ, a suburban site ZH, and a remote
and forested site at JFM.
Sulfate and Ammonium were found to have a singly peaked distribution from
0.44-1.0μm at all sites over the entire sampling period in this study, and accounted for
57.5-99 % of the daily-average total aerosol mass. Aqueous-phase reactions were
found to be an essential factor to the formation of droplet sulfate. In addition, we
found significant secondary processing and enhancement due to meteorological
drivers which were wetter or allowed for a longer residence time.
A bi-modal distribution was found for nitrate, with a droplet mode in 0.44-1.0μm,
indicating that it was formed under heavily polluted conditions or through similar
secondary aerosol processing. On the other hand, nitrate had a significant fraction in
the coarse mode in ZH and JFM during the wet season, where it accounted for about
40% of total mass. In this case, we found that the mass size distribution of nitrate was
likely attributed with chloride depletion, with almost complete chloride depletion
found in ZH and JFM during the wet season. Additionally, relative humidity was an
important consideration in chloride depletion under relatively lower relative humidity,
conditions, further leading to the increase of coarse mode nitrate.
OC and EC showed a broad peak at 0.25-1.0μm in GZ and ZH, consistent with
significant local sources, from urbanization, transport, residential, and shipping
sources. Furthermore, under less heavily polluted conditions, OC was found to have a
bi-modal distribution in JFM, with important contributions from secondary particle



formation in the fine mode and potential biological aerosol in the coarse mode
particles.
Additionally, they were shown to have broad peaks, and a significantly different
ratio, raising the likelihood of a mixing of the local emissions with emissions
transported long-range from biomass burning in Southeast Asia. These conditions
were further supported by large amount of potassium found jointly with the aerosol.
An in-depth analysis of the meteorology, and remotely sensed Fire and AOD
properties, in conjunction with a variance maximizing technique, provided further
evidence to help us validate this assumption. It is clear that there was a significant
impact on GZ and ZH from fires sources from Thailand, Laos, and Vietnam, as well
as possible long-range transport of urban emissions from the urban megacity of Hanoi
in Vietnam. The combination of local formation and long-range transport played a
significant role on the variation of particles chemical compositions.
Overall, we found that the size distribution and formation of aerosols greatly
depend on emissions, location, and in-situ processing, especially aqueous-phase
reactions. Strong local formation and long-range-transport of both urban pollution
from GZ and of biomass burning from Southeast Asia all were observed to influence
the size distribution of chemical components across all of the areas studies. On the
other hand, the interaction between sea salt aerosols and anthropogenic pollutants
showed significant effects at coastal locations and play an important role in the
deterioration of the air quality in Southern China under high relative humidity
conditions during the wet season.




**Acknowledgments**
This work was supported by National Science Fund for Outstanding Young Scholars
(41425020), China Special Fund for Meteorological Research in the Public Interest
(GYHY201406031), National Science & Technology Pillar Program
(2014BAC21B02) and Specialized Research Fund for the Doctoral Program of
Higher Education in China (2013380004115009). The authors would especially like
to thank Prof. Guenter Engling at National Tsing Hua University for helping to
chemical analysis at the laboratory.

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





**Figures**

**Figure 1.** Location of sampling sites in Southern China: GZ (Guangzhou), ZH (Zhuhai), and JFM (Jianfeng Mountain).

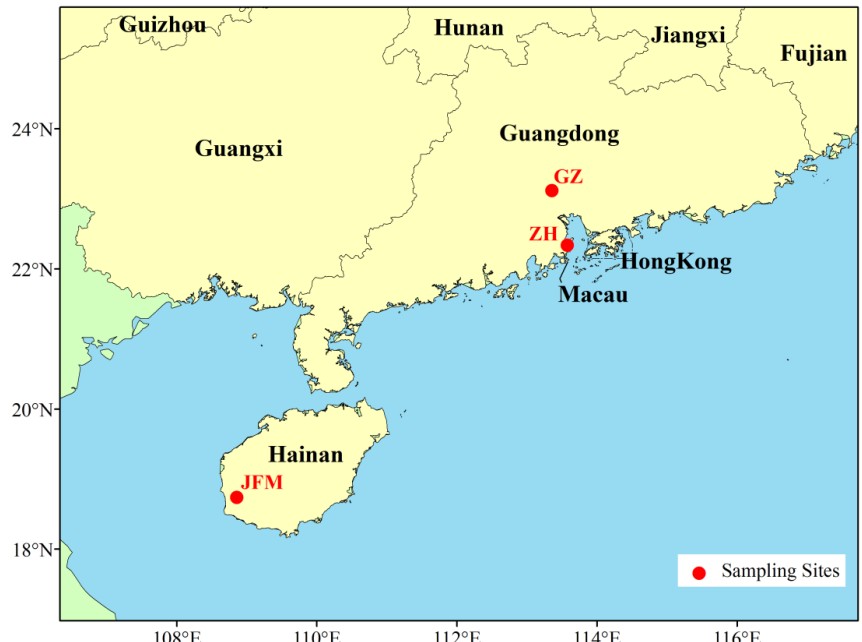





**Figure 2.** The mass size distribution of major compositions ($SO_4^{2-}$, $NO_3^-$, $NH_4^+$, OC
and EC) at the three sites during study period

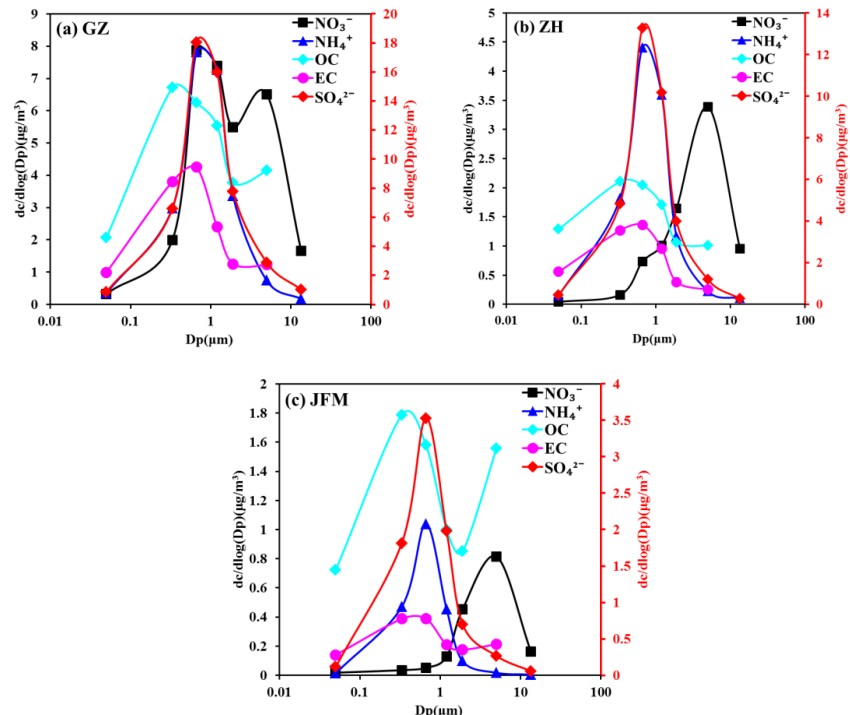




**Figure 3.** Case study on 8[th] May. in GZ ((a)The cloud fraction over Southern China;
(b)Distribution of simulated average 2 m relative humidity at 05:00-07:00 LT; (c) The
time series of observational visibility, wind speed, relative humidity and the
depression of dew point (time resolution was 30mins); (d) The time series of
monitoredmean $SO_2$ during 2009-2010 and $SO_2$ on 8[th] May ;(e) Distribution of
simulated average cloud; (f) The time-height distribution of simulated cloud water
mixing ratio on 8[th] May; (g-h) The time-height of simulated Aitken and accumulation
mode $SO_4^{2-}$)

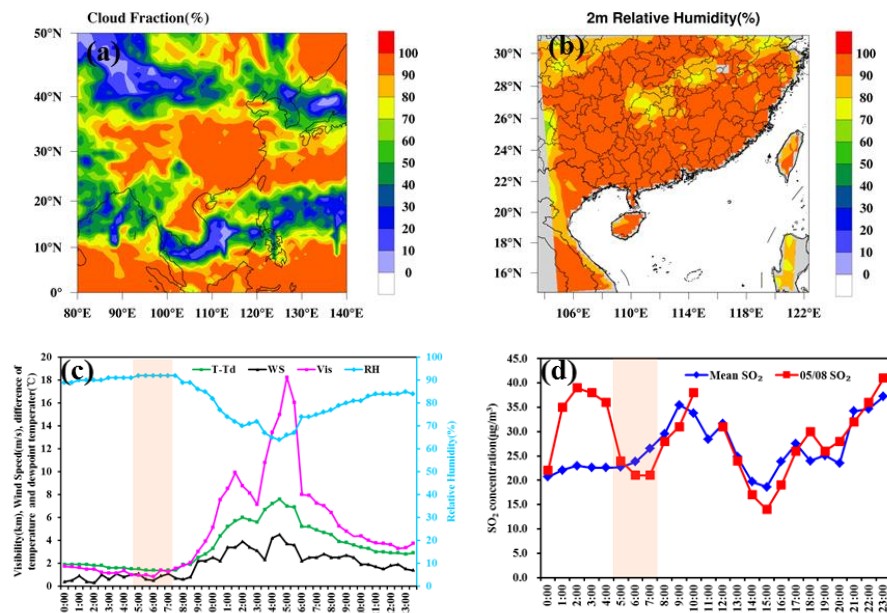



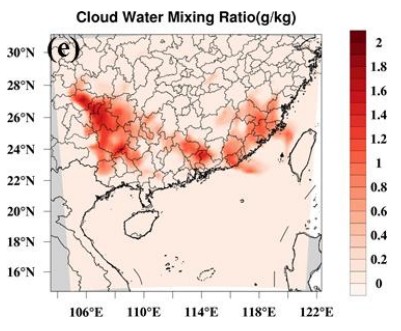

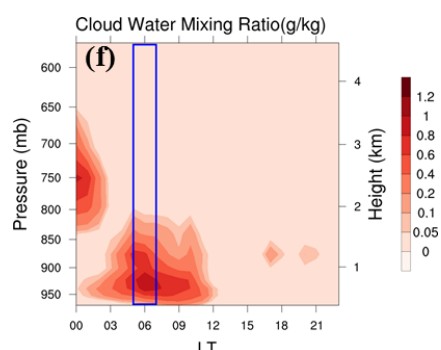

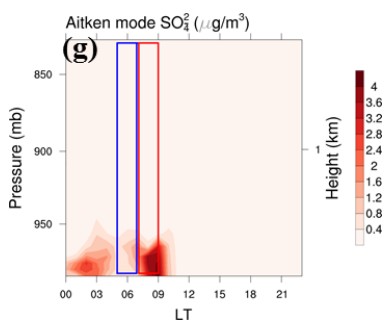

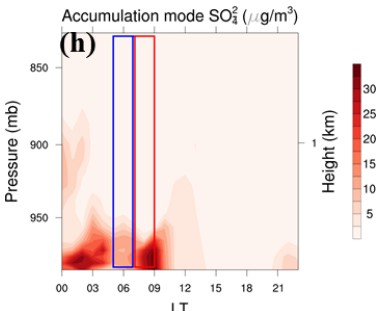





**Figure 4.** The mass size distribution of Na$^+$ and Cl$^-$ at the three sites

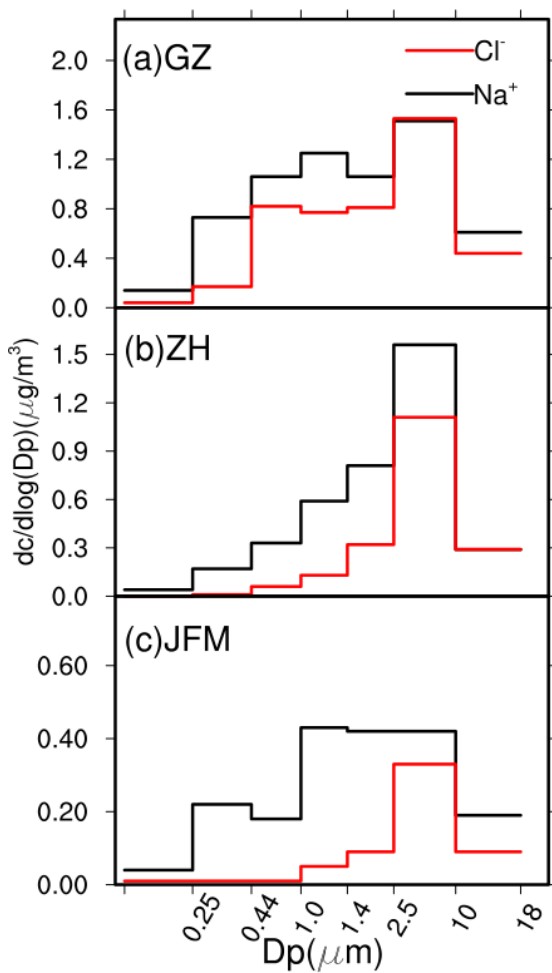






**Figure 5.** The mass size distribution of percentage of chloride depletion at the three
sites

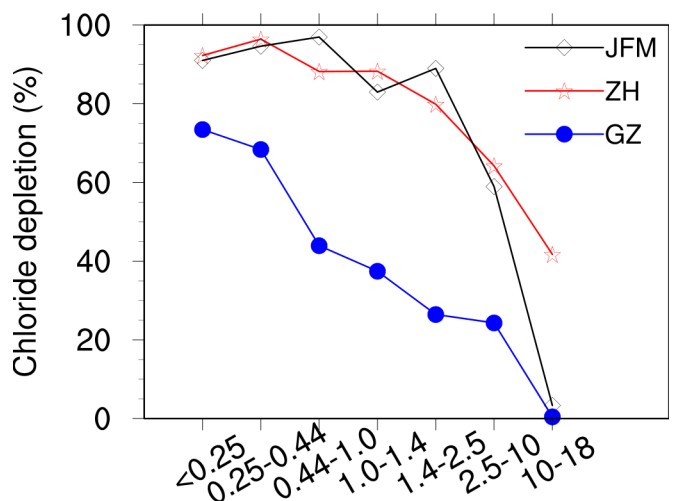





**Figure 6.** Case study on 12$^{th}$ Jun. in GZ ( (a-b) The time series of SOR and NOR; (c-d)
The mass size distribution of $SO_4^{2-}$ and $NO_3^-$; (e-f) FLEXPART-WRF total column
residence times over the last 72h arriving in GZ on 12$^{th}$ Jun. at 100m and 1000m)

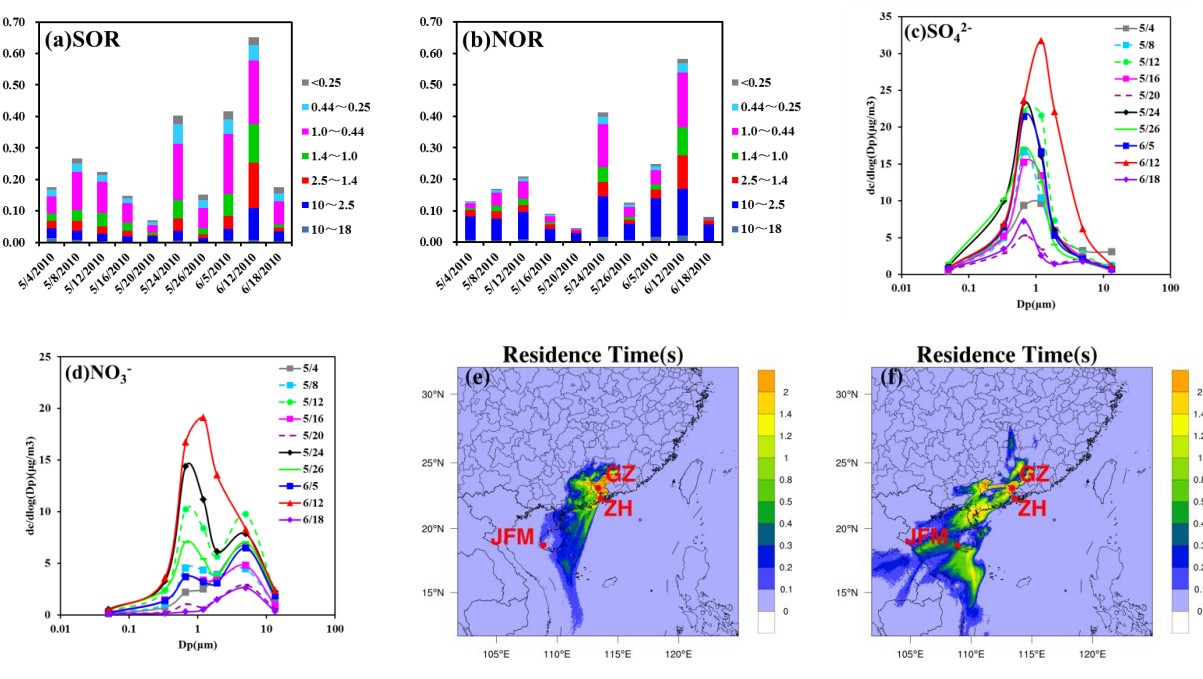




**Figure 7.** Case study on $1^{st}$, $3^{rd}$ and $5^{th}$ Jun. in JFM ((a-b) FLEXPART-WRF total
column residence times on over the last 72h arriving in JFM on $1^{st}$ Jun. at 100m and
1000m; (c-d) and (e-f) same at (a-b) but on $3^{rd}$ and $5^{th}$Jun. respectively; (g-h) The
mass size distribution of $SO_4^{2-}$ and $NO_3^-$)

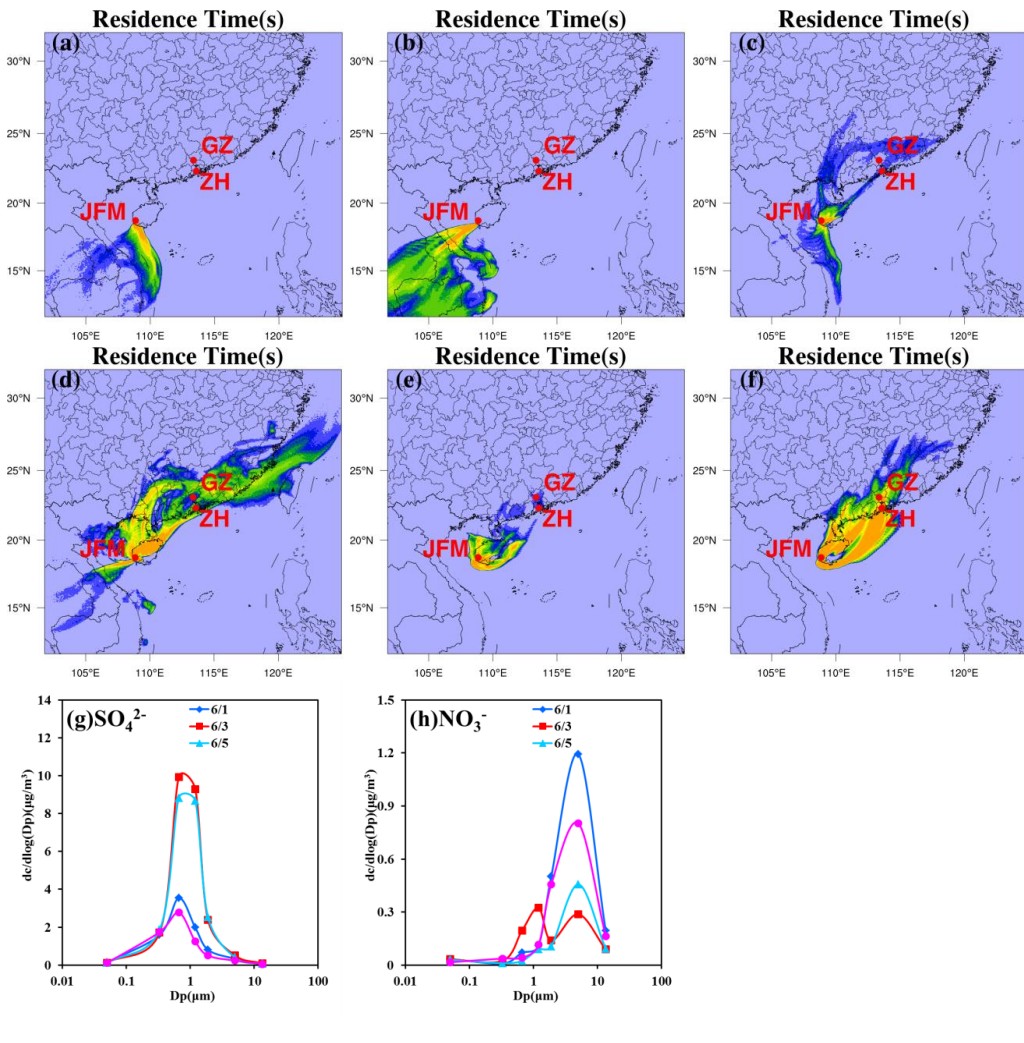








**Figure 8.** Spatially averaged/aggregated statistics of (a) MODIS Fire numbers (Count)
and (b) Fire Radiative Power (FRP) over Southeast Asia for May and June 2010. The
statistics represent the respective Count [total number of burning 1kmx1km pixels]
and average FRP [W/m$^2$ per 1kmx1km pixel] over the whole of Southeast Asia and
the specific regions where the AOD (as an indicator for smoke) has its highest levels
of variability: EOF1 and EOF2.

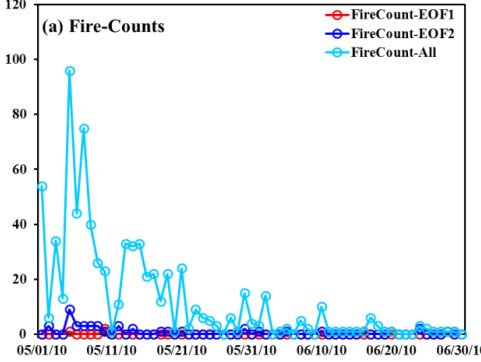
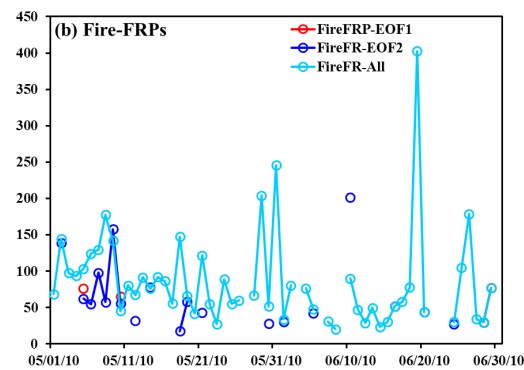






**Figure 9.** Average spatial distribution of the (a) mean and (b) standard deviation of
daily MODIS AOD from May 1$^{st}$ through June 30$^{th}$ 2010

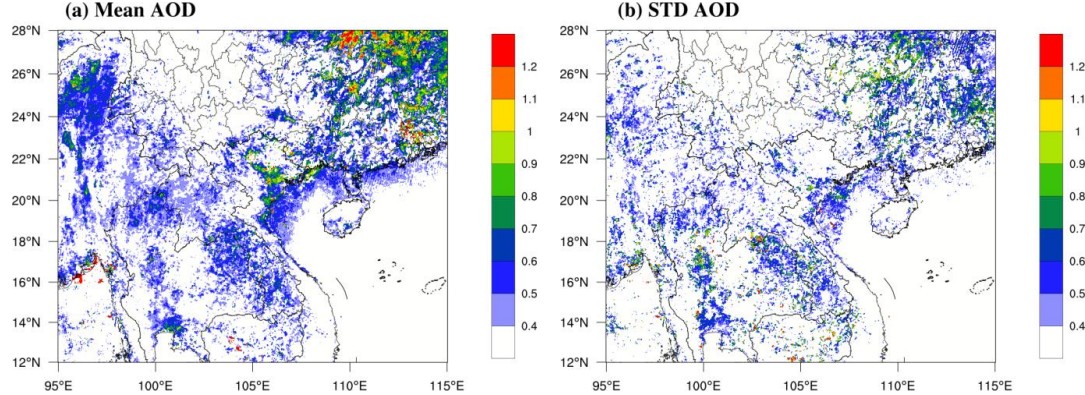






**Figure10**. The time-varying statistics of the AOD averaged over the first two EOFs of
the AOD (reflecting the regions most impacted by AOD variance or smoke from fires)
and the average $K^+$ concentration and average ratio of OC/EC in the three sites.

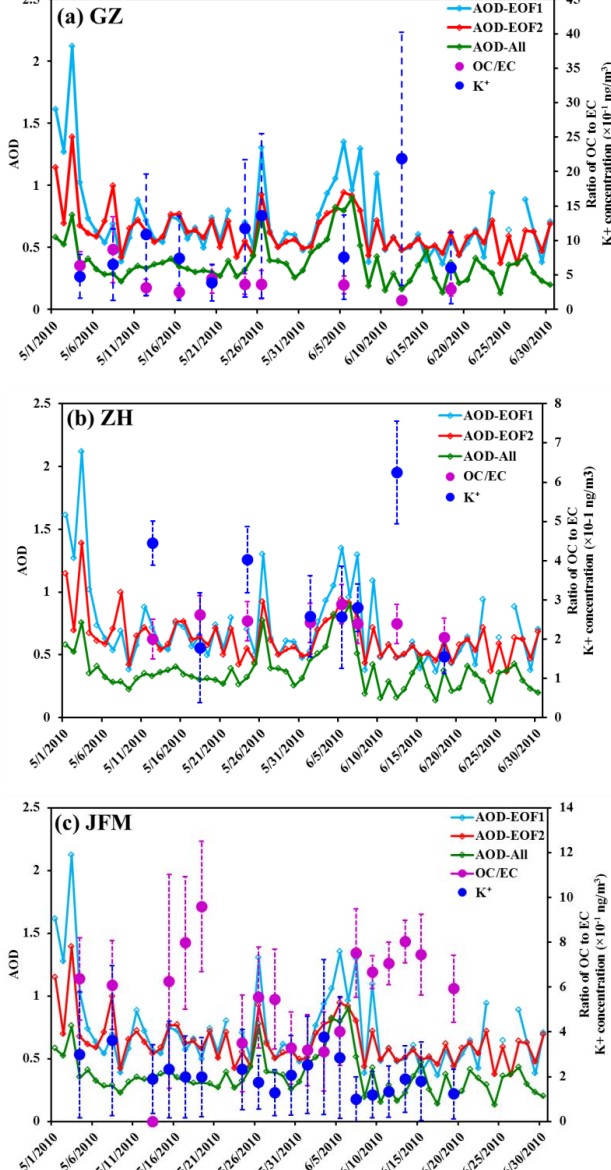






**Tables**
**Table 1.** Average concentration and standard deviation [ μg m$^{-3}$] of chemical
components in the given size-resolved particles (and their percentage of PM$_{10}$) at the
three sites during the 2010 wet season.

| Site | Size | Sum of measured species | SO$_4^{2-}$ | NO$_3^-$ | NH$_4^+$ | OC | EC |
|------|------|------|------|------|------|------|------|
| GZ | PM$_{1.0}$ | 24.4±10.9 | 8.0±3.1 (60.2) | 3.0±2.4 (34.5) | 3.4±1.7 (64.2) | 5.5±2.0 (57.9) | 2.9±2.6 (72.5) |
| | PM$_{2.5}$ | 34.9±17.3 | 11.7±5.2 (88.0) | 5.0±4.0 (57.5) | 4.9±2.9 (92.5) | 7.2±2.7 (75.8) | 3.4±3.2 (85.0) |
| | PM$_{10}$ | 46.7±20.6 | 13.3±5.8 | 8.7±5.2 | 5.3±3.1 | 9.5±3.7 | 4±3.8 |
| ZH | PM$_{1.0}$ | 12.9±4.5 | 6.3±2.1 (66.3) | 0.3±0.3 (10.3) | 2.2±0.8 (71.0) | 2.4±1.1 (66.7) | 1.3±0.8 (76.5) |
| | PM$_{2.5}$ | 18.1±6.8 | 8.8±3.2 (92.6) | 0.9±0.8 (31.0) | 3.0±1.2 (96.8) | 3.0±1.5 (83.3) | 1.5±0.9 (88.2) |
| | PM$_{10}$ | 23.7±7.3 | 9.5±3.4 | 2.9±1.1 | 3.1±1.3 | 3.6±1.9 | 1.7±1.0 |
| JFM | PM$_{1.0}$ | 4.4±1.6 | 1.8±1.0 (75.0) | 0.1±0.1 (16.7) | 0.5±0.3 (83.3) | 1.5±0.7 (57.7) | 0.3±0.2 (60.0) |
| | PM$_{2.5}$ | 5.8±2.3 | 2.2±1.5 (91.7) | 0.2±0.1 (33.3) | 0.6±0.5 (99.0) | 1.8±0.8 (69.2) | 0.4±0.2 (80.0) |
| | PM$_{10}$ | 8.0±2.6 | 2.4±1.5 | 0.6±0.3 | 0.6±0.5 | 2.6±1.1 | 0.5±0.3 |



**Table 2.** Statistical parameters of samples with air masses from ocean

| Site | Date | Droplet mode sulfate ($\mu g\ m^{-3}$) | Percentage of sulfate in droplet mode (%) | T (°C) | RH (%) | P (hPa) | WS (m s$^{-1}$) | Low Cloud cover (%) |
|------|------|------|------|------|------|------|------|------|
| GZ | 2010/5/8 | 7.4 | 61 | 27.5 | 82.0 | 997.1 | 1.9 | 70 |
|    | 2010/5/12 | 11.1 | 65 | 25.0 | 77.5 | 1002.9 | 1.5 | 60 |
| ZH | 2010/5/12 | 9.5 | 67 | 24.9 | 83.0 | 1006.1 | 3.4 | 70 |
|    | 2010/6/1 | 6.8 | 67 | 24.8 | 80.0 | 1002.0 | 5.1 | 70 |
| JFM | 2010/5/4 | 2.2 | 64 | 22.0 | 83.0 | 916.9 | 1.0 | 70 |
|     | 2010/5/13 | 2.5 | 67 | 23.7 | 75.8 | 918.3 | 1.8 | 70 |
