# Peer review of "Properties of aerosols and formation mechanisms"

_Atmospheric Chemistry and Physics, 2016_

## Referee Comment (RC1) · Anonymous Referee #1 · 17 Jun 2016

It is a pleasure to review the manuscript "Properties of aerosols and formation mechanisms over southern China during the monsoon season" by Chen et al. This manuscript addresses an important science question: what are the characteristics of size distribution and formation of atmospheric aerosols in southern China and how they are affected by local and long-range transport? Given the heavy PM concentrations in that region, answering this question has practical implications for public health. This study makes diligent use of a unique in-situ dataset, modeling, and remote-sensing products, investigates various physical and chemical mechanisms of aerosol (including secondary) formation and evolution, and provides some new insights into this scientific issue. The paper is comprehensive in its scope, well organized and well written, and the research

quality is high. I suggest to accepting this manuscript after the authors clarify the following points, which are mostly minor concerns and editorial changes:

- Line 150, the model required the use of measured: did you have all those measurements from your site observations? - Line 180, use of GDAS: why did you use higher resolution WRF/Chem meteorological fields in place of GDAS? - Line 187: the sentence is confusing. It looks like you use WRF simulated fields as input to FLEXPART, which contradicts with the previous statement of using GDAS? - Line 200: Which two sites are referring to here? There are three sites listed in Table 1. - Line 203-208: very nice analysis! - Line 287: something is missing the sentence "far-upwind urbanization and biomass burning". Urbanization cannot be transported. - Line 304,"the sites": which sites? - The quality of some figures are acceptable but not very good. For instance: o Figs 3c&d: the figure labels need to be improved. It is hard to read. o Fig 4: again the x-axis label is hard to read. o Fig 6 labels need to be improved. - Line 417: a nice example of using backward trajectories to test your hypothesis! - Section 3.6: can authors elaborate on the MODIS Fire products? Are these daily products or twice a day or 8-day?

Editorial suggestions:

- Line 92: replace "The data is" by "those data were" - Line 94: insert "of size distribution" between "characteristics" and "in" - Lin 98-99: something is missing in "and the impacts mixing sea-salt and urban pollutants" - Line 103: it would be good to show photos of these sites to give an idea of site environments - Line 110: replace "The first site was set at (23.12°N, 113.36°E), on" by "The first site (23.12°N, 113.36°E), is located on" - Line 111: replace "an urban mega-city" by "a mega-city" - Line 149: delete "to" - Line 170: replace "All of the data is" by "All data are" - Line 194: delete "mass" - Line 212-213: replace "no matter what the particle size was" by "irrespective of particle size" - Line 217: should "formed" be "from" - Line 219: delete "with" - Line 226: delete "of" - Line 242: replace "showing" by "shows" - Line 281: replace "high levels" by "high-level" - Line 288: replace "as talked about later" by "as discussed later

in this paper" - Line 303: replace "show" by "shown" - Line 312: replace "evidence" by "evidenced" - Line 323: replace "of the" by "shows that" - Line 489: delete the second "instead" - Line 527-529: replace "Based on specific case studies, some models of the air flow, and remote sensing, the impacts of chemistry and atmospheric transport were investigated" by "These site observations, together with model simulations and remote-sensing data, were used to investigate impacts of chemistry and atmospheric transport". - Line 547: delete "further" - Line 554: be specific on "they". What are they?

---

## Referee Comment (RC2) · Anonymous Referee #2 · 16 Aug 2016

In this study, size distributions of chemical components of aerosols were observed at three stations located in urban, suburban, and background areas. Results were thoroughly discussed from various aspects. I think this is a nice paper, but I feel the explanations are not enough or not adequate. They should be modified before this paper is accepted for publication.

A lot of previous studies are adequately referred. But, what are new findings of this study? Results of this study may be easily imaginable based on previous studies referred in this paper. The importance and significance, and the differences from previous studies should be mode insisted in the introduction.

I think one problem of this manuscript is that the overall observed data are not shown

and not discussed. Are all the data obtained for the target two months those shown in Figure S1? If so, this figure should be shown in the main text and overall explanation for them are necessary at first. If it is missing, I have an impression that only the day which are easy to add explanations are picked up for analyses.

Specific comments are as follows.

Line 58-60 Cohen and Wang (2013) appear twice.

Line 60-62 It may be difficult for some readers to understand modes listed here. It is better to briefly explain their definitions. Actually, this sentence is obvious because the four modes listed here (Aitken, condensation, droplet, and coarse modes) cover almost entire aerosols.

Line 61-62 I suppose that new particles via nucleation form in the Aitken mode, and not in the condensation mode. Do "new particles" mean those form on existing aerosols via condensation of gases?

Line 65 "Different" source types from what? What are differences?

Line 75-76 Pierson and Brachaczek should be (1998), not (1988).

Line 138 What do "6 sets" correspond to? There are 7 cut-off diameters.

Line 142-143 I could not understand why number of samples becomes these number. The sampling campaign was performed for two months. 24h sampling was performed every other day in GZ and ZH. So maximum number of samples is around 30, isn't it? 48h sampling was conducted every day in JFM. Does it mean two sampling instruments were used to obtain a sample for 48h every day? How the total number of samples in JFM becomes 140 only for two months? How many days the samples were properly collected and missing? Please add more explanations to understand overall pictures of the samples used in this study.

Line 144 Detailed information of the in-lab chemical analytical techniques is described

in Zhang et al. (2013a), but at least it is necessary to mention also here which species were analyzed in this study.

Line 145-147 The background literatures should be explicitly shown here, especially for the definition of droplet particles.

Line 155 Is it possible to ignore effects of other ions? Is it just because only these three ions were detected? Weren't other ions used in AIM-II model, either?

Line 176-183 Differences between HYSPLIT and FLEXPART are described, but what is a specific reason why these two models were used in this study? What are expectations from these two models in the context of this study?

Line 198-200 Does it mean that the percentages of all the samples collected at all the locations fall within such the narrow range? That is kind of incredible. Or just the averaged values shown in Table 1 fall within this range? That is nonsense. The percentages calculated for each sample should be discussed here.

Line 203 What does "the majority of individual chemical species" mean? A reason of this question is because it is unclear which species were detected in this study.

Line 207-208 Again, which are "detected chemical components"?

Line 215-216 References of "the nature of the sources" should be shown here, especially for shipping sources. Are there any references showing shipping sources are dominant around this region?

Line 217-218 I suppose mobile vehicles are not main sources for sulfate. What is "high temperature industry"?

Line 220-221 How can rapid oxidation of precursor species be a reason of differences between urban and background sites? I suppose the phase equilibrium should be also one of important reasons of differences because nitrates would move to gas phase while transported to background areas.

Line 263-265 The percentages shown here are against what?

Line 265-266 Droplet mode nitrate is formed similarly to sulfate. Does it mean that nitrate is also formed via aqueous reactions? If so, what kind of aqueous reactions? If not, why nitrate is included in the droplet mode?

Line 289 What is coarse OC with a possible source of biological aerosol? Any references?

Line 298-301 I could not understand this sentence. How the author judged the average size was small, the particles were relatively young, and indicative of new particle formation? Are all of these coming from the fact that droplet mode sulfate accounted for about two thirds of the total mass concentration of sulfate? More detailed explanations are necessary to reinforce this discussion.

Line 306-308 That is true for selected days. But, how about for days not selected? Low cloud cover 60-70% and higher relative humidity were observed only for the days selected here?

Line 335 Are the words "Accumulation mode" and "condensation mode" used for the same meaning?

Line 352 Is it possible to judge that fine mode chloride and sodium are coming from sea salts? Are there any anthropogenic sources of chloride and sodium in the fine mode? If chloride and sodium in the fine mode are emitted separately from difference sources from sea salts, discussions on chloride depletion in the fine mode in this paragraph is not appropriate.

Line 362 What does "calculated ammonium" mean? How was it calculated?

Line 381 What is another important non-linear effect? It is unreasonable to discuss reasons of percentage differences only based on humidity. A lot of other factors like emission sources on pathways should be considered.

Line 403 Please add the definitions of Sulfur Oxidation Ratio and Nitrogen Oxidation Ratio, and their importance in the context of this study.

Line 439 I cannot understand discussions around here. Why can discussions in this paragraph be a reason of long-range transport? As mentioned in the line 425-426, wind speed was very low. Isn't it possible to explain high concentrations and aging under stagnant air around urban area? Do the discussions in this paragraph enable to clearly distinguish effects of stable air and long-range transport?

Line 485 Is this paragraph saying that MODIS fire hotspots are not useful to see effects of biomass burning?

Line 494 The EOF technique may be useful, but it means that it is better than the MODIS fire hotspots discussed in the previous paragraph? What is a specific reason?

Figure 2 Please specify which species use the left and right Y-Axes.

Figure 4 Why do these figures look different from other species shown in other figures? They should be consistent.

[Figure]

---

## Author Comment (AC1) · 26 Sep 2016

**Response to Referee#1:**

We would like to thank the reviewer for the careful evaluations and positive comments on our paper, which improved the paper so much. We have revised the manuscript according to the reviewer's detailed comments. Please find the responses to the reviewers.

Reviewer's comments are in plain face.

Author responses are in blue color.

Changes in the manuscript are in red color.

**Comments to the Author:**

It is a pleasure to review the manuscript "Properties of aerosols and formation mechanisms over southern China during the monsoon season" by Chen et al. This manuscript addresses an important science question: what are the characteristics of size distribution and formation of atmospheric aerosols in southern China and how they are affected by local and long-range transport? Given the heavy PM concentrations in that region, answering this question has practical implications for public health. This study makes diligent use of a unique in-situ dataset, modeling, and remote-sensing products, investigates various physical and chemical mechanisms of aerosol (including secondary) formation and evolution, and provides some new insights into this scientific issue. The paper is comprehensive in its scope, well organized and well written, and the research quality is high. I suggest to accepting this manuscript after the authors clarify the following points, which are mostly minor concerns and editorial changes:

- Line 150, the model required the use of measured: did you have all those measurements from your site observations?

Response:

Yes, all of these required parameters were observed at these three sites, including ambient temperature, relative humidity, and the concentration of sulfate, nitrate and ammonium. We have added more information about chemical analysis in line 151-156:

'The mass concentrations of six cations ($Na^+$, $NH_4^+$, $K^+$, $Ca^{2+}$, $Mg^{2+}$, and $Ca^{2+}$) and seven anions ($F^-$, $Cl^-$, $NO_2^-$, $Br^-$, $SO_4^{2-}$, $NO_3^-$ and $PO_4^-$) were analyzed using an ion chromatography (ICS-3000, DIONEX. Thermal Optical Transmittance (TOT) technique was employed to analyze the quartz filter samples to determine the mass concentrations of organic carbon (OC) and elemental carbon (EC) by the use of Sunset Laboratory OCEC Carbon Aerosol Analyzer.'

- Line 180, use of GDAS: why did you use higher resolution WRF/Chem

meteorological fields in place of GDAS?

Response:

WRF model could provide higher resolution meteorological data, further to improve the dispersion simulations and lead to an overall better simulation (Stohl, 1998). In addition, a novel convective scheme has been added in the FLEXPART –WRF (Brioude et al., 2013), which also could improve the model simulation, especially for finer scale applications. So meteorological fields provided by WRF instead of GDAS were used in this study. We have added an explanation in line 205-209:

' The application of FLEXPART –WRF with a novel convective scheme being added improves the dispersion simulations and results in an overall better simulation, especially for finer scale applications (Brioude et al., 2013; Stohl, 1998). In this case, the region was modeled with a spatial resolution of 27×27 km and a temporal resolution of 1 hour.'

- Line 187: the sentence is confusing. It looks like you use WRF simulated fields as input to FLEXPART, which contradicts with the previous statement of using GDAS?

Response:

Just as the reviewer said, WRF simulation provided meteorological fields for ELEXPAET, while GDAS was applied by HYSPLIT. So we have deleted this sentence and just kept the previous statement in line 201-205:

'HYSPLIT uses single air parcels to compute trajectories with the use of Global Data Assimilation System (GDAS, 1°×1°) as input data. FLEXPART, on the other hand, uses a larger number of air parcels to compute trajectories based on the meteorological predictions provided by mesoscale model WRF.'

- Line 200: Which two sites are referring to here? There are three sites listed in Table 1.

Response:

We are sorry for our imprecise statement. Actually the range of [0.52, 0.55] and [0.72, 0.76] is the range for three sites not only for two sites. We have re-written this part to avoid confusion in line 226-228:

'In terms of the mass size distribution, the percentage of $PM_{1.0}$ to $PM_{10}$ was 60.2%, 66.3% and 75.0%, and $PM_{2.5}$ to $PM_{10}$ was 88.0%, 92.6%, 91.7% in GZ, ZH and JFM, respectively.'

- Line 203-208: very nice analysis!

Response: Thanks so much for the review's acceptance.

- Line 287: something is missing the sentence "far-upwind urbanization and biomass burning". Urbanization cannot be transported.

Response:

Thanks for the reviewer's reminder, we have added more information in line 316-319:

'We investigated these days and find that emissions that long-range transported from far-upwind areas with highly urbanization or with the existence of biomass burning are responsible, as discussed later in this paper.'

- Line 304,"the sites":which sites? - The quality of some figures are acceptable but not very good. For instance: o Figs 3c&d: the figure labels need to be improved. It is hard to read. o Fig 4:again the x-axis label is hard to read. o Fig 6 labels need to be improved.

Response:

Thanks for pointing out this issue. The sites represent the three sites. We have clarified it in line 339-341:

'Additionally, it was determined that during these times at the three sites there was an abnormally high amount of low cloud cover 60-70% and a relatively higher relative humidity (75~83%) (Table 2).'

Thanks so much for the review's suggestion, we have improved and re-plotted the figures in the main text.

- Line 417:a nice example of using backward trajectories to test your hypothesis!

Response: Thanks so much for the review's acceptance.

- Section 3.6: can authors elaborate on the MODIS Fire products? Are these daily products or twice a day or 8-day?

Response:

Thanks for the reviewer's suggestion, the MODIS fire products are daily products, we have provided some information about MODIS Fire products in the section *2 Measurements and methodology* in line 184-193:

'Aerosol optical depth (AOD), fire products including Fire Radiative Power (FRP), and Fire Quality Assurance [QA] data, were obtained from the MODIS sensors aboard both the AQUA and TERRA satellites. Specifically, we obtained the Collection 6, 3km Level 2 swath product for AOD (Remer et al., 2013), and Collection 5.1, 1km Level 2 swath products for FRP and QA (Giglio et al., 2006). The Collection 5.1 active fire products are daily products and have been improved based on the previous collection 5.0 products. All of the data are cloud-screened, with AOD data being computed using different algorithms over land and water, and the fire data using 19

different channels for quality assurance. We only accept values for FRP and Fire Count where the QA is at least 90%.'

And in line 550-555:

'MODIS fire hotspots are not very useful in wet and tropical regions. Since MODIS fire hotspots are obstructed by both clouds and high levels of aerosols in the atmosphere, both of which are found associated with tropical forest fires. Additionally, due to the highly wet ground surface, a significant amount of the fires may low temperature and therefore not observable using the MODIS sensors (Cohen, 2014, Giglio et al., 2006; Yu et al., 2015)'

**Editorial suggestions:**

- Line 92: replace "The data is" by "those data were"

Response:

Thanks and we have modified it.

- Line 94: insert "of size distribution" between "characteristics" and "in"

Response:

Thanks and we have added  it

- Lin 98-99: something is missing in "and the impacts mixing sea-salt and urban pollutants"

Response:

Thanks for reminding and we have added more information in line 106-107.

'and the impacts mixing sea-salt and urban pollutants on the characteristics of size distributioin .'

- Line 103: it would be good to show photos of these sites to give an idea of site environments

Response:

Thanks for the suggestion, we have added photos of these sites in Fig. 1

[Figure]

**Figure 1.** Location of sampling sites in Southern China: GZ (Guangzhou), ZH (Zhuhai), and JFM (Jianfeng Mountain) and their surrounding environments

- Line 110: replace "The first site was set at (23.12N, 113.36E), on" by "The first site (23.12N, 113.36E), is located on"

Response:

Thanks and we have modified it.

- Line 111: replace "an urban mega-city" by "a mega-city"

Response:

Thanks and we have modified it.

- Line 149: delete "to"

Response:

Thanks and we have modified it.

- Line 170: replace "All of the data is" by "All data are"

Response:

Thanks and we have modified it.

- Line 194: delete"mass"

Response:

Thanks and we have modified it.

- Line 212-213: replace "no matter what the particle size was" by "irrespective of particle size"

Response:

Thanks and we have modified it.

- Line 217: should "formed" be "from"

Response:

Thanks and we have modified it.

- Line 219: delete "with"

Response:

Thanks and we have modified it.

– Line 226: delete "of"

Response:

Thanks and we have modified it.

- Line 242: replace "showing" by "shows"

Response:

Thanks and we have modified it.

- Line 281: replace "high levels" by "high-level"

Response:

Thanks and we have modified it.

- Line 288: replace "as talked about later" by "as discussed later in this paper"

Response:

Thanks and we have modified it.

- Line 303: replace "show" by "shown"

Response:

Thanks and we have modified it.

- Line 312: replace "evidence" by "evidenced"

Response:

Thanks and we have modified it.

- Line 323: replace "of the" by "shows that"

Response:

Thanks and we have modified it.

- Line 489: delete the second "instead"

Response:

Thanks and we have modified it.

- Line 527-529: replace "Based on specific case studies, some models of the air flow, and remote sensing, the impacts of chemistry and atmospheric transport were investigated" by "These site observations, together with model simulations and Remote-sensing data, were used to investigate impacts of chemistry and atmospheric transport".

Response:

Thanks and we have modified it.

- Line 547: delete "further"

Response:

Thanks and we have modified it.

- Line 554: be specific on "they". What are they?

Response:

Thanks and 'they' represent 'OC and EC', we have clarified it.

**Reference:**

Brioude, J., Arnold, D., Stohl, A., Cassiani, M., Morton, D., Seibert, P., Angevine, W., Evan, S., Dingwell, A., Fast, J.D., Easter, R.C., Pisso, I., Burkhart, J. and Wotawa, G.: The Lagrangian particle dispersion model FLEXPART-WRF version 3.1, Geoscientific Model Development, 6, 1889-1904, doi:10.5194/gmd-6-1889-2013, 2013.

Stohl, A.: Validation of the lagrangian particle dispersion model FLEXPART against large-scale tracer experiment data, Atmospheric Environment, 32, 4245-4264, doi:10.1016/S1352-2310(98)00184-8, 1998.

Yu, C., Chen, L.F., Li, S.S., Tao, J.H. and Su, L.: Estimating Biomass Burned Areas from Multispectral Dataset Detected by Multiple-Satellite, Spectroscopy and Spectral Analysis, 35, 739-745, doi:10.3964/j.issn.1000-0593(2015)03-0739-07, 2015.

---

## Author Comment (AC2) · 26 Sep 2016

**Response to Referee#2:**

We would like to thank the reviewer for the careful evaluations and positive comments on our paper, which improved the paper so much. We have revised the manuscript according to the reviewer's detailed comments. Please find the responses to the reviewers.

Reviewer's comments are in plain face.

Author responses are in blue color.

Changes in the manuscript are in red color.

**Comments to the Author:**

In this study, size distributions of chemical components of aerosols were observed at three stations located in urban, suburban, and background areas. Results were thoroughly discussed from various aspects. I think this is a nice paper, but I feel the explanations are not enough or not adequate. They should be modified before this paper is accepted for publication.

A lot of previous studies are adequately referred. But, what are new findings of this study? Results of this study may be easily imaginable based on previous studies referred in this paper. The importance and significance, and the differences from previous studies should be made insisted in the introduction.

I think one problem of this manuscript is that the overall observed data are not shown and not discussed. Are all the data obtained for the target two months those shown in Figure S1? If so, this figure should be shown in the main text and overall explanation for them are necessary at first. If it is missing, I have an impression that only the day which are easy to add explanations are picked up for analyses.

Response:

(1). A series of studies about the mass size distribution of aerosol chemical components have conducted at a specific site over Southern China during the past decade (Lan et al., 2011; Liu et al., 2008; Yang and Wenig, 2009; Zhang et al., 2015). Compared with the previous studies, this paper presents a unique combination of analytical and measurement techniques. We use measurements of chemical properties and size distribution conducted at three different functional sites, coupled with multiple modeling results, and reprocess remote sensing products using statistical methods, all in tandem with each other, which is not commonly found in other studies. Furthermore, we test our approach in Southern China, which is one of the regions of the world with the most complex meteorology, coming under the influence of the Monsoon, with shifting winds from Continental and Oceanic sources. Additionally, the season tested is a transition period, during which there were significant meteorological contributions from both remote Continental sources as well as oceanic sources. On top of this, Southern China has a combination of high temperature and relative humidity, strong radiative flux, and high oxidative capacity, leading to the

promotion of significant secondary aerosol formation.This paper will provide detailed information on size-resolved aerosol chemical components and discussion on the formation mechanism in a typical region and period. We have emphasized the importance and significance in the introduction in line 84-99:

'A series of studies about the mass size distribution of aerosol chemical components have conducted at a specific site over Southern China during the past decade (Lan et al., 2011; Liu et al., 2008; Yang and Wenig, 2009; Zhang et al., 2015). Compared with the previous studies, this paper presents a unique combination of analytical and measurement techniques. We use measurements of chemical properties and size distribution conducted at three different functional sites, coupled with multiple modeling results, and reprocess remote sensing products using statistical methods, all in tandem with each other, which is not commonly found in other studies. Furthermore, we test our approach in Southern China, which is one of the regions of the world with the most complex meteorology, coming under the influence of the Monsoon, with shifting winds from continental and oceanic sources. Additionally, the season tested is a transition period, during which there were significant meteorological contributions from both remote continental sources as well as oceanic sources. On top of this, Southern China has a combination of high temperature and relative humidity, strong radiative flux, and high oxidative capacity, leading to the promotion of significant secondary aerosol formation.'

(2). The overall observed data for the target two months were shown in Figure S1, which has now been moved into the paper as Figure 2. Additionally, all of the data obtained for the target months is displayed in line 145-150:' A total of 10, 8 and 20 sets of size-segregated particle samples were collected in GZ, ZH and JFM, respectively during the periods of May and June in 2010 (shown in Figure 2). A single set of sample collection lasted for approximately 24h in GZ and ZH, and 48h in JFM. Since the aerosol concentration was relatively low in remote JFM site, we extended the sampling time as long as 48h to allow the chemical components to be detected.'and line 217-223:' The time series of $PM_{18}$ chemical compositions at the three sites during the sampling period were shown in Figure 2. The average concentration and standard deviation of $PM_{18}$ was 47.8±20.8, 24.3±12.1 and 8.1±2.7 µg m$^{-3}$ in GZ, ZH and JFM, respectively. The mean and range of $PM_{18}$ in highly urban GZ was both higher and wider than in suburban ZH and rural JFM, with the respective ranges being 23.3~93.7, 13.3~35.1, 4.7~14.3µg m$^{-3}$ in the three sites. Maximum concentration was found both on 12$^{th}$ Jun. in GZ and ZH, while it was on 3$^{rd}$ Jun. in JFM (to be discussed later).'

**Specific comments are as follows:**

-Line 58-60 Cohen and Wang (2013) appear twice.

Response:

Thanks for pointing out that and we have deleted the repeated reference.

-Line 60-62 It may be difficult for some readers to understand modes listed here. It is better to briefly explain their definitions. Actually, this sentence is obvious because the four modes listed here (Aitken, condensation, droplet, and coarse modes) cover almost entire aerosols.

Response:

Thanks for the suggestion and we have provided the definition for the four modes in line 59-61:

'In the environment, the most important aerosol processes occur over the Aitken (<0.1 μm), condensation (~0.1-0.5 μm), droplet (~0.5-2.0μm), and coarse (>2.0 μm) size modes (Seinfeld and Pandis, 2006)'

Just as the reviewer mentioned, the sentence is obvious because the four modes cover almost entire aerosols, what we want to emphasize is the different processes occurred on different modes as shown in line 61-63:

'new particles are formed in the Aitken mode via condensational growth and coagulation of nucleation mode particles, and droplet mode particles are produced by in-cloud processing or aqueous reactions.'

Line 61-62 I suppose that new particles via the nucleation form in the Aitken mode, and not in the condensation mode. Do "new particles" mean those form on existing aerosols via condensation of gases?

Response:

Thanks for pointing out this issue. Just as the reviewer suggested, the new particles are formed in the Aitken mode via nucleation formation. Here, new particles mean that condensational growth and coagulation of nucleation mode particles. We have modified it in line 61-62:

'new particles are formed in the Aitken mode via condensational growth and coagulation of nucleation mode particles'

Line 65 "Different" source types from what? What are differences?

Response:

Coarse mode aerosols usually originate from natural or mechanically produced anthropogenic sources, for example, sea spray, soil dust, dust storm, active biological aerosol (pollen, spores), etc. We have clarified this sentence in line 65-68:

'On the other hand, coarse mode aerosols usually come from very different sources than smaller aerosols. For example, natural sources such as sea spray, dust, soil, and active biological aerosols are unique and therefore provide further information about the aerosol distribution at a given location'

Line 75-76 Pierson and Brachaczek should be (1998), not (1988).

Response:

Thanks for pointing out this mistake and we have amended it.

Line 138 What do "6 sets" correspond to? There are 7 cut-off diameters.

Response:

Thanks for pointing out this clerical error and it should be 7 sets. We have corrected it in the paper.

Line 142-143 I could not understand why number of samples becomes these number. The sampling campaign was performed for two months. 24h sampling was performed every other day in GZ and ZH. So maximum number of samples is around 30, isn't it? 48h sampling was conducted every day in JFM. Does it mean two sampling instruments were used to obtain a sample for 48h every day? How the total number of samples in JFM becomes 140 only for two months? How many days the samples were properly collected and missing? Please add more explanations to understand overall pictures of the samples used in this study.

Response:

We are sorry for our imprecise statement. The sampling was performed on a specific day, but not every other day, the specific date was shown in Figure 2. Only one instrument was operated in JFM since the concentration of aerosol was relatively lower, so we extended the sampling time as long as 48h to allow for sufficient connection of the chemical components so that they could be detected. We have clarified it in line 143-150:

'To attain size-segregated particle samples, a 6-stage High Flow Impactor (MSP) with an airflow rate of 100 L min$^{-1}$ was employed, with cutoff diameters ($D_p$) of 18 (inlet), 10, 2.5, 1.4, 1.0, 0.44 and 0.25 μm. A total of 10, 8 and 20 sets of size-segregated particle samples were collected in GZ, ZH and JFM, respectively, during the periods of May to June in 2010 (shown in Figure 2). A single set of sample collection lasted for approximately 24h in GZ and ZH, and 48h in JFM. Since the aerosol concentration was relatively low in JFM, we extended the sampling time as long as 48h to make the chemical components detected.'

Line 144 Detailed information of the in-lab chemical analytical techniques is described in Zhang et al. (2013a), but at least it is necessary to mention also here which species were analyzed in this study.

Response:

We agree with the comment and we have added more information about chemical analysis in line 151-156:

'The mass concentrations of six cations ($Na^+$, $NH_4^+$, $K^+$, $Ca^{2+}$, $Mg^{2+}$, and $Ca^{2+}$) and seven anions ($F^-$, $Cl^-$, $NO_2^-$, $Br^-$, $SO_4^{2-}$, $NO_3^-$ and $PO_4^-$) were analyzed using an ion chromatography (ICS-3000, DIONEX. Thermal Optical Transmittance (TOT) technique was employed to analyze the quartz filter samples to determine the mass concentrations of organic carbon (OC) and elemental carbon (EC) by the use of Sunset Laboratory OCEC Carbon Aerosol Analyzer.'

Line 145-147 The background literatures should be explicitly shown here, especially for the definition of droplet particles.

Response:

Thanks for the suggestion and we have added more information in line 159-163:

'To be consistent with the background literature (4 modes include Aitken (<0.1 μm), condensation (~0.1-0.5 μm), droplet (~0.5-2.0 μm), and coarse (>2.0 μm)) (Seinfeld and Pandis, 2006), and the constraints of the size bins measured in this study, we implement 2.5 μm as the cut-off size to separate fine and coarse particles, and the size bins from 0.44-1.4 μm was defined as droplet particles in this study.'

Line 155 Is it possible to ignore effects of other ions? Is it just because only these three ions were detected? Weren't other ions used in AIM-II model, either?

Response:

Actually, the mass concentrations of six cations ($Na^+$, $NH_4^+$, $K^+$, $Ca^{2+}$, $Mg^{2+}$, and $Ca^{2+}$) and seven anions ($F^-$, $Cl^-$, $NO_2^-$, $Br^-$, $SO_4^{2-}$, $NO_3^-$ and $PO_4^-$) were analyzed in this study, as we have added the chemical analysis in section *2 Measurements and methodology*. The possible error would be introduced due to the excluding of other ions (e.g. $K^+$, $Na^+$, $Cl^-$), but which exerted only a minor influence on the estimation of aerosol acidity due to their lower concentration. And, Yao et al. (2006) found that AIM provides the most accurate prediction compared with other models, like ISORROPIA and SCAPE2. We have added an explanation in line 163-169: 'Although we were not able to directly measure aerosol water content, given its importance for the study here, we instead estimated the amount by the use of E-AIM model II (Clegg et al., 1998), as it provides the most accurate prediction compared with other models, like ISORROPIA and SCAPE2 (Yao et al., 2006). The input parameters of the E-AIM model II are tempreture, relative humidity, strong acidity ($H^+$), molar contentrations of $NH_4^+$, $SO_4^{2-}$ and $NO_3^-$ ions (Clegg et al., 1998)' and in line 172-174: 'The calculation of strong acidity would introduce possible errors due to the exclusion of other ions( e.g. $K^+$, $Na^+$, $Cl^-$), but which only exerted a minor influence on the estimation of aerosol acidity due to their lower concentration (Yao et al., 2006).'

Line 176-183 Differences between HYSPLIT and FLEXPART are described, but what is a specific reason why these two models were used in this study? What are expectations from these two models in the context of this study?

Response:

HYSPLIT and FLEXPART were applied to determine the origin of air masses. Furthermore, FLEXPART could identify the relative importance of source region that affected the receptor visually. We have added some information to illustrate our expectation in line 196-201:

'Two Lagrangian particle dispersion models, the Hybrid Single Particle Lagrangian Integrated Trajectory (HYSPLIT) (Draxler and Hes, 1998) and FLEXPART coupled with The Weather and Research and Forecasting (WRF) model (FLEXPART –WRF) (Stohl et al., 1998; Brioude et al., 2013) were applied to determine the origin of air masses in this study, Compared with HYSPLIT, FLEXPART can identify the relative importance of source region that affected the receptor visually.'

Line 198-200 Does it mean that the percentages of all the samples collected at all the locations fall within such the narrow range? That is kind of incredible. Or just the averaged values shown in Table 1 fall within this range? That is nonsense. The percentages calculated for each sample should be discussed here.

Response:

Thanks for the comment. The percentage is the averaged values at the three sites shown in Table 1. We have re-written this part to avoid confusion in line 224-229:

'Table 1 listed the average concentration of chemical components in the given size-resolved particle ($PM_{1.0}$, $PM_{2.5}$ and $PM_{10}$) and their percentage of $PM_{10}$ at the three sites. In terms of the mass size distribution, the percentage of $PM_{1.0}$ to $PM_{10}$ was 60.2%, 66.3% and 75.0%, and $PM_{2.5}$ to $PM_{10}$ was 88.0%, 92.6%, 91.7% in GZ, ZH and JFM, respectively. When considered as a whole, it is the smaller sized particles that dominate the aerosol loading at all three of these sites.'

Line 203 What does "the majority of individual chemical species" mean? A reason of this question is because it is unclear which species were detected in this study.

Response:

Thanks for pointing out this issue. We have added more information on aerosol chemical components in line 151-156:'The mass concentrations of six cations ($Na^+$, $NH_4^+$, $K^+$, $Ca^{2+}$, $Mg^{2+}$, and $Ca^{2+}$) and seven anions ($F^-$, $Cl^-$, $NO_2^-$, $Br^-$, $SO_4^{2-}$, $NO_3^-$ and $PO_4^-$) were analyzed using an ion chromatography (ICS-3000, DIONEX. Thermal Optical Transmittance (TOT) technique was employed to analyze the quartz filter samples to determine the mass concentrations of organic carbon (OC) and elemental carbon (EC) by the use of Sunset Laboratory OCEC Carbon Aerosol Analyzer.' and we have clarified it in line 229-231:'Looking at the data on a species-by-species level,

most of chemical components were concentrated in fine mode particles, which contributed at least 57% to $PM_{2.5}$'

Line 207-208 Again, which are "detected chemical components"?

Response:

Thanks for pointing out this issue. The detected chemical components including six cations ($Na^+$, $NH_4^+$, $K^+$, $Ca^{2+}$, $Mg^{2+}$, and $Ca^{2+}$),  seven anions ($F^-$, $Cl^-$, $NO_2^-$, $Br^-$, $SO_4^{2-}$, $NO_3^-$ and $PO_4^-$) and carbonaceous aerosol (OC and EC). We have added more information on aerosol chemical components in line 151-156:

'The mass concentrations of six cations ($Na^+$, $NH_4^+$, $K^+$, $Ca^{2+}$, $Mg^{2+}$, and $Ca^{2+}$) and seven anions ($F^-$, $Cl^-$, $NO_2^-$, $Br^-$, $SO_4^{2-}$, $NO_3^-$ and $PO_4^-$) were analyzed using an ion chromatography (ICS-3000, DIONEX. Thermal Optical Transmittance (TOT) technique was employed to analyze the quartz filter samples to determine the mass concentrations of organic carbon (OC) and elemental carbon (EC) by the use of Sunset Laboratory OCEC Carbon Aerosol Analyzer.'

Line 215-216 References of "the nature of the sources" should be shown here, especially for shipping sources. Are there any references showing shipping sources are dominant around this region?

Response:

Thanks for pointing out this issue and we have added the related references in line 241-245:

'These findings are consistent with the nature of the sources of sulfur from industrial and  power plant (Zheng et al., 2009). In addition, shipping source was becoming increasingly vital for $SO_2$ emission with an increment of 12% per year in this region (Lu et al., 2013; Zhou et al, 2016)'

Line 217-218 I suppose mobile vehicles are not main sources for sulfate. What is "high temperature industry"?

Response:

We agree with the comment that mobile vehicle are not the main sources of sulfate. High temperature industry reprensent industry and power plant. So we have re-written this sentence to avoid confusion in line 246-247:

 'Nitrate, mainly formed from the oxidation of NOx emitted by mobile vehicles and power plants, showed a remarkable difference between urban and background site'

Line 220-221 How can rapid oxidation of precursor species be a reason of differences between urban and background sites? I suppose the phase equilibrium should be also

one of important reasons of differences because nitrates would move to gas phase while transported to background areas.

Response:

We agree with that phase equilibrium is one of the important reasons in background areas.However, rapid oxidation of precursors is the main source for nirate in urban area. So we have changed the expression in line 249-253:

'This is consistent with its more rapid oxidation of its abundant precursor species, especially so in the urban atmosphere (Cohen et al., 2011). In addition, phase equilibrium was another important reason for the discrepancy since nitrate would tend to exist as gas phase while transported to background areas (Seinfeld and Pandis, 2006).'

Line 263-265 The percentages shown here are against what?

Response:

Thanks for pointing out this issue. We have deleted this sentence.

Line 265-266 Droplet mode nitrate is formed similarly to sulfate. Does it mean that nitrate is also formed via aqueous reactions? If so, what kind of aqueous reactions? If not, why nitrate is included in the droplet mode?

Response:

Thanks for pointing out this issue. The formation of droplet mode nitrate is unlike sulfate. Droplet mode nitrate was dominated by the heterogeneous aqueous reaction of gaseous nitric acid ($HNO_3$) and ammonia ($NH_3$) on the wet surfaces of pre-existing aerosols with ammonia-rich environment, otherwise by heterogeneous hydrolysis of $N_2O_5$ on the pre-existing aerosols with ammonia-poor conditions. The dissociation equilibrium of $NH_4NO_3$ highly depends on temperature and humidity (Stelson and Seinfeld, 1982). We have re-written these sentences in line 293-298:

'Droplet mode nitrate was dominated by the heterogeneous aqueous reaction of gaseous nitric acid ($HNO_3$) and ammonia ($NH_3$) on the wet surfaces of pre-existing aerosols within ammonia-rich environment, otherwise by heterogeneous hydrolysis of $N_2O_5$ on the pre-existing aerosols within ammonia-poor conditions. The dissociation equilibrium of $NH_4NO_3$ highly depends on temperature and humidity (Stelson and Seinfeld, 1982)'

Line 289 What is coarse OC with a possible source of biological aerosol? Any references?

Response:

The possible source for coarse OC would be active biological aerosol, for example,

pollen, spores, plant fragment. We have added more information in line 319-323:

'Additionally, there is some coarse mode OC present in JFM, suggesting a possible source of biological aerosol (e.g. pollen, spores and plant fragment) (Heald and Spracklen, 2009; Seinfeld and Pandis, 2006; Zhang et al., 2015), which is consistent with the large amounts of vegetation present in that region (Zhang et al., 2015).'

Line 298-301 I could not understand this sentence. How the author judged the average size was small, the particles were relatively young, and indicative of new particle formation? Are all of these coming from the fact that droplet mode sulfate accounted for about two thirds of the total mass concentration of sulfate? More detailed explanations are necessary to reinforce this discussion.

Response:

Thank you for your suggestion. After careful analysis, we admit that we could not draw this conclusion just based on the fact that droplet mode sulfate accounted for about two thirds of the total mass concentration of sulfate. Therefore, we have deleted this sentence to avoid the confusion.

Line 306-308 That is true for selected days. But, how about for days not selected? Low cloud cover 60-70% and higher relative humidity were observed only for the days selected here?

Response:

Thanks for pointing out this issue. In order to study the aqueous-phase reaction of droplet mode sulfate, cases with the air masses came from continent were excluded to avoid the effect of transported pollutants on the concentration of sulfate. Then two cases with a concentration of droplet sulfate above the mean plus one standard deviation were selected as typical cases in each site. It was found that the selected cases happened under the conditions of higher relative humidity and low cloud cover. Lower cloud cover (60-70%) and higher relative humidity were also observed for other non-selected days, which also shared with higher percentage (60% above) of sulfate in droplet mode sulfate to total sulfate but with relatively lower absolute concentrations. Therefore cased with extremely high droplet mode sulfate were selected to study, which were more obvious to observe. We have added more description in line 327-334:

'In order to study the aqueous-phase reaction of droplet mode sulfate, cases with the air masses came from continent were excluded to avoid the effect of transported pollutants on the concentration of sulfate. Then two cases with a concentration of droplet sulfate above the mean plus one standard deviation were selected as typical cases in each site (8$^{th}$ and 12$^{th}$ May in GZ, 12$^{th}$ May and 1$^{st}$ Jun. in ZH, and 4$^{th}$ and 13$^{th}$ May 2010 in JFM), which were more obvious to observe to investigate the effect of aqueous-phase reaction in the formation of droplet mode sulfate (blue shade in

Figure 2).'

Line 335 Are the words "Accumulation mode" and "condensation mode" used for the same meaning?

Response:

The diameter for accumulation mode particles was ~0.1-1μm as for WRF/Chem model, while the diameter for condensation and droplet mode was ~0.1-0.5 and ~0.5-2.0μm. So accumulation mode and condensation mode are not exactly the same, but the diameter for accumulation mode is between the range for condensation and droplet mode.We have clarified it in line 370-372:

'Simulation of these conditions using WRF/Chem indicates that the rapid growth of both Aitken and accumulation (~0.1-1μm) mode sulfate started at 07:00 LT and peaked at 08:00-09:00 LT'

Line 352 Is it possible to judge that fine mode chloride and sodium are coming from sea salts? Are there any anthropogenic sources of chloride and sodium in the fine mode? If chloride and sodium in the fine mode are emitted separately from difference sources from sea salts, discussions on chloride depletion in the fine mode in this paragraph is not appropriate.

Response:

Thanks so much for pointing out this issue. Fine mode $Na^+$ and $Cl^-$ probably came from combustion sources, e.g. biomass burning and coal combustion (Wang et al., 2005), but the contributions were insignificant since the magnitude of $Na^+$ and $Cl^-$ from combustion sources is many orders of magnitude smaller than oceanic sources. Furthermore, if the biomass burning source were significant, it would clearly also show up in terms of the $K^+$ and BC/OC ratio, as explained later in the section *3.6 Quantifying the impacts of fires*.

We totally agree with the reviewer's comment that the anthropogenic emissions would also affect chloride depletion in fine mode particles. So in the revised version, we introduced an indicator, concentration of chloride depletion ($[Cl_{dep}]$), to simply remove the possible effect of non-sea salt emissions. Therefore, samples with possible non-sea salt sources were excluded from analysis to avoid the effect of non-sea salt emission on chloride depletion in fine mode particles. We have clarified it in line 379-399:

'The mass size distribution of $Cl^-$ and $Na^+$ showed a similar pattern to nitrate at the three sites, peaking in coarse mode particles (Figure 5 (a-c)) with an average percentage of 43%, 62% and 43% for coarse mode $Na^+$, 53%, 76% and 74% for coarse mode $Cl^-$ in GZ, ZH and JFM, respectively, suggesting the main sea salt sources. $Na^+$ and $Cl^-$ shown a bi-modal distribution in GZ, illustrating the combustion

emissions, e.g. biomass burning or coal combustion for fine mode $Na^+$ and $Cl^-$ (Wang et al., 2005), but the contributions were insignificant since the magnitude of $Na^+$ and $Cl^-$ from combustion sources is many orders of magnitude smaller than oceanic sources. Furthermore, if the biomass burning source were significant, it would clearly also show up in terms of the $K^+$ and BC/OC ratio, as explained later in the section *3.6 Quantifying the impacts of fires*.

The concentration and percentage of chloride depletion ($[Cl_{dep}]$ and $\%Cl_{dep}$) were calculated using Eq. (2-3), where $[Cl_{meas}^-]$ and $[Na_{meas}^+]$ are the measured molar concentrations of $Cl^-$ and $Na^+$, respectively; 1.174 was the molar ratio of $Cl^-$ to $Na^+$ in sea water (Yao et al., 2003b).

$$[Cl_{dep}]=1.174[Na_{meas}^+]-[Cl_{meas}] \tag{2}$$

$$\%Cl_{dep}=\frac{1.174[Na_{meas}^+]-[Cl_{meas}]}{1.174[Na_{meas}^+]}*100\% \tag{3}$$

The positive value of $[Cl_{dep}]$ represents chloride depletion, otherwise means the chloride enrichment, suggesting additional sources was existed for $Cl^-$ excluding sea salt. Therefore, samples with negative $[Cl_{dep}]$ were excluded from analysis to avoid the effect of non-sea salt emission on chloride depletion.'

Line 362 What does "calculated ammonium" mean? How was it calculated?

Response:

Calculated ammonium was equal to $2*[nss\text{-}SO_4^{2-}]+[NO_3^-]$, where$[nss\text{-}SO_4^{2-}]$ and $[NO_3^-]$ reprents the molar concentarion of non-sea-salt $SO_4^{2-}$ (i.e., $[nss\text{-}SO_4^{2-}]=[SO_4^{2-}]-0.14\times[Cl^-]$ ) and $NO_3^-$. We have added related information in line 413-416:

'Calculated ammonium was equal to $2*[nss\text{-}SO_4^{2-}]+[NO_3^-]$, where$[nss\text{-}SO_4^{2-}]$ and $[NO_3^-]$ reprents the molar concentarion of non-sea-salt $SO_4^{2-}$ (i.e., $[nss\text{-}SO_4^{2-}]=[SO_4^{2-}]-0.14\times[Cl^-]$ ) and $NO_3^-$ (Huang et al., 2004)'

Line 381 What is another important non-linear effect? It is unreasonable to discuss reasons of percentage differences only based on humidity. A lot of other factors like emission sources on pathways should be considered.

Response:

Thanks for pointing out this issue. There are indeed three important sources of non-linearity, but we do not mention them all. First, chemical/aqueous phase non-linearity. Second, meteorological non-linearity. When the wind direction changes from ocean, to land, or to long-range transport, that the chemical compositions are different, as

well as the time under which in-situ chemistry could occur. Third, emissions non-linearity. When there are fire sources, the emissions are significantly different from when there are non-fire sources, and this extends to the $NO_2$ emissions (and hence NOx).

We estimated the chloride depletion in ZH and JFM when the air masses came from the ocean or continent. Actually, it didn't show much difference for %$Cl_{dep}$ no matter where the air masses came from. We agree with the reviewer's comment that antoher factors like emission sources also need to be considered. So drawing the conclusion that 'suggesting another important non-linear effect between maritime aerosols and anthropogenic NOx' just based on the percentage difference would be inadiquate. Therefore we have deleted this paragraph to make the paper more robust.

Line 403 Please add the definitions of Sulfur Oxidation Ratio and Nitrogen Oxidation Ratio, and their importance in the context of this study.

Response:

The Sulfur Oxidation Ratio (SOR) and Nitrogen Oxidation Ratio (NOR), which were used to indicate the degree of transformation of secondary aerosol (Wang *et al.*, 2005). We have added related information in line 454-461:
'The Sulfur Oxidation Ratio (SOR) and Nitrogen Oxidation Ratio (NOR) are applied to indicate the degree of oxidation of $SO_2$ and $NO_2$ precursor gases (Wang *et al.*, 2005). The equations for SOR and NOR are calculated as SOR=$n$-$SO_4^{2-}$/($n$-$SO_4^{2-}$+$n$-$SO_2$) andNOR=$n$-$NO_3^-$/($n$-$NO_3^-$+$n$-$NO_2$), where $n$-$SO_4^{2-}$and $n$-$NO_3^-$are the molar concentrations of particulate $SO_4^{2-}$ and $NO_3^-$ and $n$-$SO_2$ and $n$-$NO_2$ are the molar concentrations of the precursor gases $SO_2$ and $NO_2$. SOR and NOR was also the highest On $12^{th}$ June at the range of 0.44-1.0 μm in GZ with the value of 0.20 and 0.17, respectively'

Line 439 I cannot understand discussions around here. Why can discussions in this paragraph be a reason of long-range transport? As mentioned in the line 425-426, wind speed was very low. Isn't it possible to explain high concentrations and aging under stagnant air around urban area? Do the discussions in this paragraph enable to clearly distinguish effects of stable air and long-range transport?

Response:

Thanks for pointing out this issue. Discussion in this paragraph was only one aspect of long-range transport, the following discussions were additional evidences to support this conclusion. We clearly explain this more precisely as follows:
First, HYSPLIT and FLEXPART-WRF showed that the air flow was mostly from Southeast Asia at levels over the boundary layer, and hence had undergone long-range transport. Second, the windspeed near the surface suddenly became very low. Therefore, there had been a rapid change in the windspeed. Based on conservation laws for air mass, it would be excepted for there to be some reasonable amount of mixing of the air vertically. This was consistent with the finding that some of the air

which had undergone long-range transport would have mixed into the surface region. Third, the chemical concentrations of BC, OC, and $K^+$ were all elevated, $Na^+$ and $Cl^-$ size distribution were peaked in the 1.0-1.44 μm size range, and they were bi-modal distribution in ZH, which were also consistent with a significant fire contribution. Fourth, the time-gap between when the air parcels left Southeast Asia and arrived in Guangzhou and Zhuhai, correspond very well with times during which the EOFs over Southeast Asia showed a significantly large amount of smoke (Figure 10), which would be discussed in section **3.6. *Quantifying the impacts of fires***.

We make clear this in the text in line 495-515:

'Except for in-situ formation, long-range transport was another impact factor. First, HYSPLIT (Take GZ for example, Figure S2(a)) and FLEXPART-WRF (Figure 6(f)) showed that the air flow was mostly from Southeast Asia at levels over the boundary layer ((Figure 6(f)), and hence had undergone long-range transport. Second, the windspeed near the surface suddenly became very low. Therefore, there had been a rapid change in the windspeed. Based on conservation laws for air mass, it would be excepted for there to be some reasonable amount of mixing of the air vertically. This was consistent with the finding that some of the air which has undergone long-range transport would have mixed into the surface region. Furthermore, the ratio of OC to EC concentrations was the minimum measured values on 12$^{th}$ June, with a mean ratio of 1.32 and 2.39 in GZ and ZH, respectively. Also, OC showed a bi-modal distribution, although predominantly in the fine mode while EC mostly peaked at fine mode particles (Take GZ for example, Figure S2 (g-h)), indicating that the organic aerosol was mostly primary, as would be expected from large fire sources. Additionally, the $K^+$ concentration on 12$^{th}$ June was about 2-3 times higher than that of mean value measured in GZ and ZH (Take GZ for example, Figure 10(a-b), and Figure S1(i)). $Na^+$ and $Cl^-$ size distribution were found to be uni-modal distribution in GZ, where they peaked in the 1.0-1.44μm size range. $Na^+$ and $Cl^-$ were bi-modal distribution in ZH on 12$^{th}$ Jun. (figures not showed here). All of these findings above, including the time of the year and the location, are consistent with the long-range transported biomass burning from Southeast Asia (Cohen, 2014).'

Line 485 Is this paragraph saying that MODIS fire hotspots are not useful to see effects of biomass burning?

Response:

We do not state that in general MODIS fire hotspots are not useful. In fact, they have been shown to be very useful in many dry areas and in many temperate and arctic areas. This is why they are commonly used. However, our results show that MODIS fire hotspots are not very useful in wet and tropical regions. This has been published before, such as Cohen, 2014, Giglio et al., 2006 and Yu et al., 2015. MODIS fire hotspots are obstructed by both clouds and high levels of aerosols in the atmosphere, both of which are found associated with tropical forest fires. Additionally, due to the highly wet ground surface, a significant amount of the fires may low temperature and therefore not observable using the MODIS sensors.We make clear this in the text in

line 550-555:

'This result showed that the MODIS fire hotspots are not very useful in wet and tropical regions. MODIS fire hotspots are obstructed by both clouds and high levels of aerosols in the atmosphere, both of which are found associated with tropical forest fires. Additionally, due to the highly wet ground surface, a significant amount of the fires may low temperature and therefore not observable using the MODIS sensors (Cohen, 2014, Giglio et al., 2006; Yu et al., 2015)'

Line 494 The EOF technique may be useful, but it means that it is better than the MODIS fire hotspots discussed in the previous paragraph? What is a specific reason?

Response:

There are some physical and mathematical reasons for this. First of all, observing aerosols is significantly easier and more precise. Since they are measured using variables in the visible and infrared, their measurement is more robust than the hotspot products, which are only in the infrared. There are many articles which show that the aerosol errors are roughly 10% of their total value, whereas for fires, they are significantly higher (Morisette et al., 2005a, 2005b; Levy et al., 2007, 2010; Remer et al., 2007).

From a mathematical sense, we are interested in looking at contributions which are significant. Fire hotspots are effectively point measurements, and as such are not spatially robust. Therefore, it is hard to tell what type of significant impact they have on the atmosphere in general. Given the uncertainties in such precise meteorology, it is highly probable that a point measurement may not convert precisely into an inverse method of atmospheric transport. Whereas AOD is an area measurement, and is a continuous measurement, since the aerosols transport and spread. Therefore, if a significant signal exists, it is far easier to track and transport, at the scale of the inverse meteorological methods used here.

We have added more information in line 558-563:

'Since MODIS fire hotspots are effectively point measurements, and as such are not spatially robust, while AOD are continuous and more easier to be observed,  and provides more precise and robust spatial information (Morisette et al., 2005a, 2005b; Levy et al., 2007, 2010; Remer et al., 2005). Therefore, if a significant signal exists, it is far easier to track and transport, at the scale of the inverse meteorological methods used here.'

Figure 2 Please specify which species use the left and right Y-Axes.

Response:

Thanks for the comment. The right Y-Axes was only used for sulfate and the left Y-Axes was for nitrate, ammonium, OC and EC. We have clarified it in line 976-977:

'The mass size distribution of major compositions ($SO_4^{2-}$, $NO_3^-$, $NH_4^+$, OC and EC) at the three sites during the study period ($SO_4^{2-}$ is plotted against the right Y-Axes)'

Figure 4 Why do these figures look different from other species shown in other figures? They should be consistent.

Response:

Thanks for the suggestion and we have re-plotted this Figure 5 in line 988-990 to make all the figures consistent.

[revised manuscript text omitted]